# Support-Set Context Matters for Bongard Problems

**Nikhil Raghuraman**                                                    *nikhilvr16@gmail.com*
*Department of Computer Science*
*Stanford University*

**Adam W. Harley**                                                       *aharley@cs.stanford.edu*
*Department of Computer Science*
*Stanford University*

**Leonidas Guibas**                                                      *guibas@cs.stanford.edu*
*Department of Computer Science*
*Stanford University*

**Reviewed on OpenReview:** *https: // openreview. net/ forum? id= kUuPUIPvJ6*

## Abstract

Current machine learning methods struggle to solve Bongard problems, which are a type of IQ test that requires deriving an abstract "concept" from a set of positive and negative "support" images, and then classifying whether or not a new query image depicts the key concept. On Bongard-HOI, a benchmark for natural-image Bongard problems, most existing methods have reached at best 69% accuracy (where chance is 50%). Low accuracy is often attributed to neural nets' lack of ability to find human-like symbolic rules. In this work, we point out that many existing methods are forfeiting accuracy due to a much simpler problem: they do not adapt image features given information contained in the support set as a whole, and rely instead on information extracted from individual supports. This is a critical issue, because the "key concept" in a typical Bongard problem can often only be distinguished using multiple positives and multiple negatives. We explore simple methods to incorporate this context and show substantial gains over prior works, leading to new state-of-the-art accuracy on Bongard-LOGO (75.3%) and Bongard-HOI (76.4%) compared to methods with equivalent vision backbone architectures and strong performance on the original Bongard problem set (60.8%). Code is available at `https://github.com/nraghuraman/bongard-context`.

## 1 Introduction

In the 1960s, Mikhail Bongard published a set of puzzles designed to highlight visual reasoning capabilities that computers lack (Bongard, 1968). The idea is to present two sets of images, where in the first set there is some abstract concept shared across all images (e.g., each image shows a triangle), and in the second set, that key concept is missing from all images (e.g., each image shows a non-triangle polygon). The goal is to name the concept in play. These puzzles can be made arbitrarily diverse and complex, making them useful intelligence tests for machines and humans alike (Hofstadter, 1979). A sample problem is shown in Figure 1.

Now – 70 years later – computer vision systems still lack the capability to reliably solve such puzzles (Mitchell, 2021). Recent work deals with simplified versions of the task, such as Bongard-LOGO (Nie et al., 2021) and Bongard-HOI (Jiang et al., 2023), where the goal is to classify whether or not a query image satisfies the concept in play (i.e., few-shot classification, as opposed to naming the abstract concept). Even in this simplified task, average accuracy has often only reached the 55-65% range (Nie et al., 2021; Jiang et al., 2023). This evaluation includes popular techniques such as contrastive learning (He et al., 2020) and meta-learning (Lee et al., 2019; Chen et al., 2021), which have seen success in other few-shot learning problems.

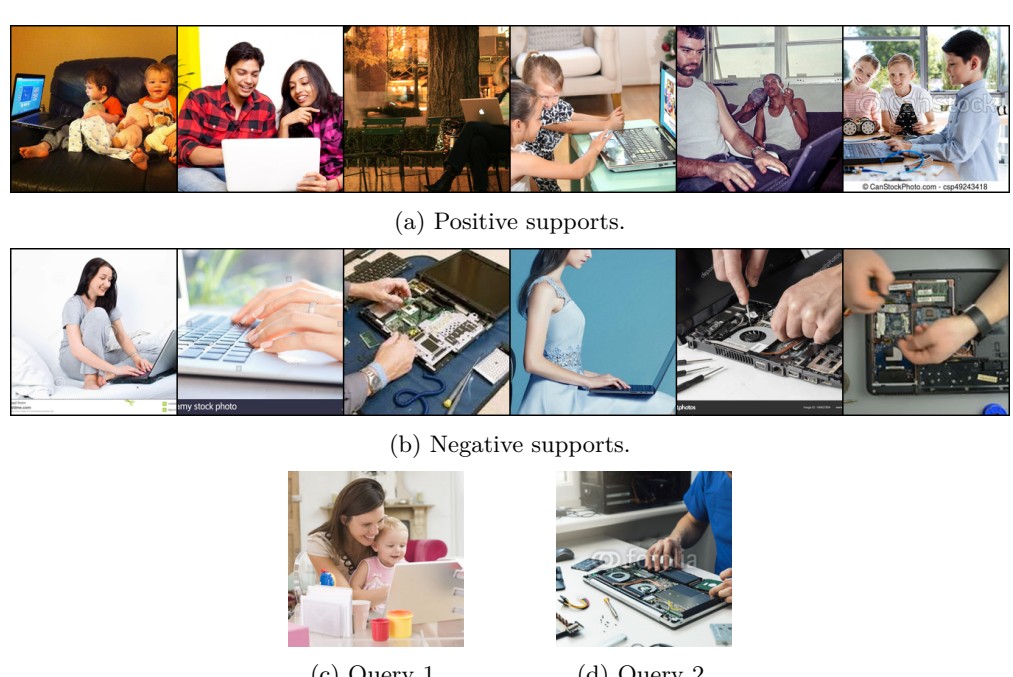

(a) Positive supports.

(b) Negative supports.

(c) Query 1.  (d) Query 2.

Figure 1: **Sample problem from Bongard-HOI.** The positives share a concept, and the negatives lack that concept. Considering all supports and one query at a time, is the query a positive or negative example of the key concept? Our model's outputs, and ground truth, are in the footnote[1].

Several recent works have pointed to low accuracy in Bongard problems as a reflection of shortcomings in the "deep learning" paradigm overall (Greff et al., 2020; Mitchell, 2021; Nie et al., 2021; Jiang et al., 2023). While this may still be an accurate assessment, we propose that neural nets' ability to solve Bongard problems has been under-estimated, due to subtle errors in setup. In our experiments, we revisit multiple baseline approaches and demonstrate that they can be improved up to 5-10 points with a very simple change: standardize the feature vectors computed from the images, using a mean and variance computed within each problem. This change has a big effect because it incorporates *context* gathered across the other examples in the problem. Incorporating set-level knowledge is a known technique in few-shot learning literature (Ye et al., 2020), though not currently used in state-of-the-art methods (Hu et al., 2022). In Bongard problems, however, using set-level knowledge is critical, because the tasks are often only solvable by using multiple supports, including both positives and negatives. We name this set-level knowledge "support-set context".

This paper has two main contributions. First, we demonstrate the effectiveness of a simple parameter-free technique to incorporate this set-level knowledge, which we call support-set standardization. Using support-set standardization, we re-evaluate the effectiveness of standard few-shot learning methods on Bongard problems, yielding a slate of new baselines with accuracies substantially higher than previously reported. Second, we explore a Transformer-based approach to learn set-level knowledge, and use explicit set-level training objectives, which force the Transformer to extract "rules" or "prototypes" from the available supports.

Our Transformer-based approach sets a new state-of-the-art on Bongard-LOGO (75.3%) and Bongard-HOI (76.4%) compared to approaches with equivalent backbone architectures, and also performs fairly well on the original Bongard problem set (60.8%).

---

[1]The concept in play is "read laptop". Our best model ("SVM-Mimic", introduced in a later section) predicts a positive score (8.7) for the first query, and a negative score (-8.3) for the second query, which is correct.

## 2 Related Work

**Puzzle solving** Visual puzzles are a useful tool for measuring progress on high-level desiderata for machine-learned models, such as few-shot concept learning, logical reasoning, "common sense", and compositionality (Greff et al., 2020). A popular area of work revolves around Raven's Progressive Matrices (Raven, 1956), which are a multiple-choice IQ test where the correct choice is indicated by a subtle pattern in "context" images, and sometimes clarified by an elimination procedure on the choices. Researchers have devised large-scale auto-generated datasets in this style (Barrett et al., 2018; Zhang et al., 2019; Spratley et al., 2020), opening the door to large-scale learning-based models, and revealing shortcomings of existing end-to-end methods. Bongard problems (Maksimov & Bongard, 1968) were designed from the outset to measure deficiencies in computer vision models (as opposed to humans), but similarly measure abstract reasoning capability (Linhares, 2000). Powerful specialized algorithms have been proposed for solving the original collection of Bongard problems (Grinberg et al., 2023; Foundalis, 2006; Depeweg et al., 2018; Sonwane et al., 2021; Youssef et al., 2022). Our own interest is not in these particular puzzles, but rather to make progress on the core abilities required to solve such puzzles in general. In line with this perspective, researchers have produced auto-generated large-scale Bongard datasets, such as Bongard-LOGO (Nie et al., 2021), where the images consist of compositions of arbitrary geometric shapes and strokes, and Bongard-HOI (Jiang et al., 2023), where the images consist of real-world photos of humans interacting with various objects. The dataset authors have benchmarked a variety of approaches, with these baselines' top accuracies reaching approximately 66% in Bongard-LOGO and Bongard-HOI, attesting to the difficulty of the task.

**Few-shot learning** Solving a Bongard puzzle essentially requires few-shot learning. In many few-shot learning settings, the target concept is indicated only by positive samples, but the concept can also be distinguished from previously-learned concepts (Miller et al., 2000; Fei-Fei et al., 2006; Lake et al., 2015). In a Bongard problem, the concept is indicated by examples *and* counter-examples, but any given concept may overlap with previous concepts in arbitrary ways. Promising methods here include meta-learning (Thrun & Pratt, 2012; Finn et al., 2017) and prototype learning (Vinyals et al., 2016; Snell et al., 2017). One approach with good results on both Bongard-LOGO (Nie et al., 2021) and Bongard-HOI (Jiang et al., 2023) is "Meta-Baseline" (Chen et al., 2021), which first learns a classifier for a set of base concepts, and then meta-learns an objective which seeks similarity between a query and its corresponding support set. Similar to this paper, Ye et al. (2020) propose a "set-to-set" approach, where support vectors are adapted using set-level context with a Transformer, and these are sent to a downstream classification algorithm. Our work has similar motivations, but we use standardization as a feature adaptation technique and use a student-teacher setup to force the Transformer to learn a robust classifier.

**Vision-language models** Contrastive Language-Image Pre-training (CLIP) (Radford et al., 2021) jointly learns an image encoder and text encoder, and trains for contrastive matching of these encodings, using an enormous private dataset of image-text pairs. An exciting result from that work is "prompt engineering", whereby a new visual concept can be acquired "zero shot" (in the sense that no visual examples are provided), by simply providing it with a short text description (or "prompt"). Numerous methods have applied CLIP to Bongard problems. Shu et al. (2022) proposed Test-time Prompt-Tuning (TPT), which performs prompt tuning on CLIP at *test time*, so as to maximally retain generalization capability to unseen data. While TPT optimizes inputs to CLIP's text encoder, BDC-Adapter (Zhang et al., 2023) trains additional small adapter networks on top of frozen features to improve visual reasoning performance. The "Augmented Queries" approach proposed by Lei et al. (2023) makes use of CLIP image and text features for data augmentation but trains a separate encoder for query classification. At the time of writing, TPT, BDC-Adapter, and Augmented Queries perform the best on Bongard-HOI, each attaining between 66% and 69% accuracy. Our proposed methods build upon CLIP similarly to methods such as BDC-Adapter, but attain much better performance as we will later show.

Large autoregressive vision-language models also can be applied toward Bongard problems. For example, GPT-4V (Achiam et al., 2023) can be adapted to the task; Zhao et al. (2023) also provide a vision-language model designed to have strong performance on multimodal in-context learning tasks, including Bongard-HOI.

By constructing multimodal prompts, they attain 74.2% accuracy on a small subset of Bongard-HOI. It is difficult to compare their performance to other methods, as they do not evaluate on the full dataset.

## 3 Method

### 3.1 Problem Definition

A Bongard problem is a type of puzzle which tests the ability to learn concepts from a limited number of visual examples. A problem consists of a set of positive sample images $\mathcal{P}_I = \{I_1^P, \dots, I_K^P\}$, a set of negative sample images $\mathcal{N}_I = \{I_1^N, \dots, I_K^N\}$, and a query image $I^Q$. Typically, the size of each set is small, for example $K = 6$. Images in $\mathcal{P}_I$ all include a specific visual concept (e.g., "read laptop" in Figure 1), and images in $\mathcal{N}_I$ do not include that concept. The positive and negative samples are also called "supports", or "the support set". The task is to classify the query image as positive or negative, using the support set as a guide.

### 3.2 Vision Backbone

Our method begins with a "vision backbone", which is a model that maps the set of input images into a set of feature vectors. These vectors will be used in the subsequent sections to attempt to solve Bongard problems.

We define an image encoder $\phi$, which maps an image $I \in \mathbb{R}^{H \times W \times 3}$ to a vector $f \in \mathbb{R}^C$. We pass every image (positives, negatives, and the query) through the encoder, yielding a set of positive features $\mathcal{P}_f = \{f_1^P, \dots, f_K^P\}$, a set of negative features $\mathcal{N}_f = \{f_1^N, \dots, f_K^N\}$, and a query feature $f^Q$.

We would like for dot products in feature space to indicate some conceptual similarity. Metric learning methods, such as contrastive learning (Chen et al., 2020a) and triplet loss functions (Schroff et al., 2015), can provide this property. For Bongard problems with natural images, we find that the visual encoder from CLIP (Radford et al., 2021), which is trained via a contrastive loss on a large dataset of Internet images, performs well. For Bongard problems with more abstract shapes (such as the line drawings in Bongard-LOGO (Nie et al., 2021)), we find that CLIP performs poorly due to the distribution shift from its training dataset, and we therefore train a ResNet (He et al., 2016) from scratch on the training set with a contrastive loss. For fairest comparison with prior works, we use the modified ResNet architecture from Nie et al. (2021) with 15 convolutional layers, which they name ResNet-15. For a pair of vectors $f_i, f_j$ where both are in $\mathcal{P}_f$, we encourage the vectors to be similar. For any pair of vectors $f_i, f_k$ where $f_i \in \mathcal{P}_f$ and $f_k \in \mathcal{N}_f$, we ask the vectors to be dissimilar. More formally, taking a positive pair of indices $(i, j)$, where $f_i, f_j \in \mathcal{P}_f$ and $i \neq j$, we define a loss $\ell_{i,j}$ as:

$$-\log \frac{\exp(\text{sim}(f_i, f_j)/\tau)}{\exp(\text{sim}(f_i, f_j)/\tau) + \sum_{f_k \in \mathcal{N}_f} \exp(\text{sim}(f_i, f_k)/\tau)}, \tag{1}$$

where $\tau$ is a temperature hyperparameter (set to 0.1) and sim is cosine similarity. This loss was also proposed by Song & Yuan (2024c) and Liu et al. (2021).

### 3.3 Simple Baseline Classifiers

In a feature space where distances indicate conceptual similarity, there are a variety of reasonable classifiers.

**$k$-nearest neighbors**   Classify the query as the majority class of its $k$ closest neighbors in the support set.

**Prototypes**   Compute the mean vector from the positives, and the mean vector from the negatives, yielding two "prototypes", and classify the query by measuring which prototype is closer. This is also called nearest class mean (Jain et al., 2000; Mensink et al., 2013).

**SVM**   Fit a max-margin hyperplane between the positive and negative supports, and classify the query using this hyperplane (Hearst et al., 1998).

### 3.4 Support-Set Context via Standardization

The simple baselines in the previous subsection actually miss important contextual information, due to the fact that the features produced by the vision backbone are all independent. Figure 1 helps illustrate the importance of considering context from multiple supports: observing that a laptop is involved in *all* supports (positive and negative), a human may effectively ignore this visual attribute and pay attention to other details to solve the problem. This motivates "in-context feature adaptation", which we define as the adaptation of $\mathcal{P}_f$, $\mathcal{N}_f$, and $f^Q$ within a single problem, using "context" contained in the support set $\mathcal{P}_f \cup \mathcal{N}_f$.

A simple way to perform in-context feature adaptation is to standardize the mean and variance across the support set, for each feature dimension independently. This would accentuate what is unique in the positives vs. the negatives if there are any similar feature dimensions prior to standardization. Taking the full set of positive and negative support features (within a problem) as $\mathcal{S}_f = \mathcal{P}_f \cup \mathcal{N}_f$, we compute the element-wise mean $\mu \in \mathbb{R}^C$ and standard deviation $\sigma \in \mathbb{R}^C$ of this set, and standardize all vectors using those statistics: $f_i = (f_i - \mu)/\sigma$. The query $f^Q$ is standardized using $\mu$ and $\sigma$ as well but does not contribute to the statistical calculation.

In principle this standardization step could be applied anywhere in the pipeline, but we find it is convenient to apply it immediately after the vision backbone. That is, after we convert the set of images to a set of feature vectors, we standardize the set, and proceed with a classifier. Although standardization using support-set statistics has been applied in the past to few-shot problems (Bronskill et al., 2020; Nichol et al., 2018; Wang et al., 2019), it is to the best of our knowledge frequently missed in state-of-the-art methods in the few-shot (e.g., Hu et al. (2022)) and Bongard literature. As we will demonstrate, this simple learning-free step improves results substantially.

### 3.5 Support-Set Context via Transformer

We next turn to the problem of attending to support-set context using a *learned* algorithm. The statistical standardization from the previous section can be seen as a learning-free method, which operates one problem at a time, with the aim of improving the success of a downstream classifier. With access to a whole dataset of problems (as is the case in Bongard-LOGO and Bongard-HOI), it should be possible to learn priors, and also absorb the classifier into the learner.

We use a Transformer encoder (Vaswani et al., 2017) to incorporate support-set context through self-attention. The main input to our Transformer encoder is the set of support vectors $f_i \in \mathcal{P}_f \cup \mathcal{N}_f$ (after statistical standardization); each support vector acts as a "token" in the Transformer. To inform the Transformer on the labels of the supports, we learn indicator vectors for "positive" and "negative", and add the corresponding indicator to each support. We additionally input one or two learnable "task" tokens (i.e., free variables in the overall optimization), to represent the classifier at the output layer.

We would like the Transformer to learn a process that converts the supports into a classifier, or "rule", taking into account not only the specific supports given, but also knowledge acquired across a training dataset. To achieve this, we propose a kind of student-teacher approach, where the "student" model must perform the same task as the "teacher" but using only partial (or noisy) information, forcing it to learn a prior. We propose to use our "simple baseline classifiers" as teachers. Specifically, we explore two variations:

1. **Prototype-Mimic:** We ask the Transformer to regress class prototypes. Specifically, we compute two ground-truth prototypes $p$ and $n$ by taking the element-wise mean of the standardized positive and negative supports, respectively. We train the Transformer with two additional learnable "task" tokens and regress the output tokens at these positions, which we name $\hat{p}$ and $\hat{n}$, to match the ground-truth $p$ and $n$ prototypes. Our loss is formulated using a cosine similarity:

$$\mathcal{L}_{\text{mimic}} = \left( 1 - \frac{\hat{p} \cdot p}{|\hat{p}||p|} \right) + \left( 1 - \frac{\hat{n} \cdot n}{|\hat{n}||n|} \right), \tag{2}$$

where $\hat{p}$, $\hat{n}$, $p$, and $n$ are vectors.

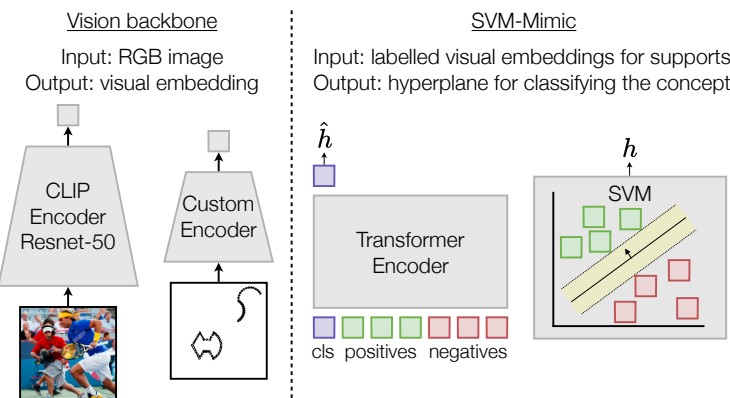

Figure 2: **Vision Backbone and SVM-Mimic.** Given "support" images labelled positive and negative, our goal is to obtain a hyperplane which can accurately classify new query images. We use a vision backbone to obtain an embedding for each support image, and then feed these in parallel to a Transformer encoder and to an SVM. Both modules output a hyperplane, and we penalize the cosine distance between them. For natural images, we use the encoder from CLIP; for geometric drawings, we train an encoder from scratch.

2. **SVM-Mimic:** We regress an SVM hyperplane computed over the standardized supports with a Transformer. Specifically, we compute a ground-truth hyperplane $h$ by fitting an SVM to the Bongard problem, which produces the coefficients and intercept of the decision boundary. We use one learnable "task" token to estimate the hyperplane, and train its output $\hat{h}$ with cosine similarity:

$$\mathcal{L}_{\text{mimic}} = 1 - \frac{\hat{h} \cdot h}{|\hat{h}||h|}, \tag{3}$$

where again $\hat{h}$ and $h$ are vectors. This approach is illustrated in Figure 2.

Crucially, we use an asymmetric student-teacher setup: the teacher (i.e., averaging or SVM) uses *all available supports* in the Bongard problem, yielding "clean" training targets. In fact, we allow the teachers to use the problem's queries $f^Q$ as additional supports to produce the strongest possible targets. The student (i.e., the Transformer) receives only a random subset of supports, forcing it to learn to complete missing information (i.e., the remaining supports and the queries in the problem). We refer to this step as "support dropout". To make the learner's task even more challenging, we use label noise: we randomly flip the positive/negative indicators for some supports (but leave the majority intact, to keep the problem well-posed). These training details encourage the Transformer to not only match the "simple baseline" teachers, but exceed them in terms of robustness to number of supports and errors in labels.

It is unconventional to use a student (in this case the Transformer) that contains many more learnable parameters than the teacher (Prototypes or SVM). However, by making the student's task harder than the teacher's task and training the Transformer across a dataset of Bongard problems, we intend for the Transformer to learn priors that generalize *across* Bongard problems and make it more robust than an SVM or Prototype baseline. We later show that the Transformer outperforms the SVM and Prototype baselines, which cannot be trained across the entire dataset of Bongard problems.

A unique aspect of our approach is that we directly supervise the learning of (ideal) prototypes, or (max-margin) rules, with a regression loss. This is instead of supervising for the correct classification of queries as Ye et al. (2020) do, with for example a cross entropy loss.

When training a ResNet backbone from scratch, we train both the backbone and the Transformer jointly by summing both loss terms. Note that it is possible to allow gradients to flow through the Transformer into the vision backbone, but for simplicity we omit this.

Table 1: **Results on Bongard-LOGO.** Error bars are obtained by evaluating three trained models over the entire test set (thousands of problems), for all methods except deterministic ones.

| Method | Enc. | Free-Form | Basic | Combinatorial | Novel | Average |
|---|---|---|---|---|---|---|
| SNAIL (Mishra et al., 2018) | RN12 | $56.3 \pm 3.5$ | $60.2 \pm 3.6$ | $60.1 \pm 3.1$ | $61.3 \pm 0.8$ | 59.5 |
| ProtoNet (Snell et al., 2017) | RN12 | $64.8 \pm 0.9$ | $72.4 \pm 0.8$ | $62.4 \pm 1.3$ | $65.4 \pm 1.2$ | 66.3 |
| MetaOptNet (Lee et al., 2019) | RN12 | $60.3 \pm 0.6$ | $71.7 \pm 2.5$ | $61.7 \pm 1.1$ | $63.3 \pm 1.9$ | 64.3 |
| ANIL (Raghu et al., 2020) | RN12 | $56.6 \pm 1.0$ | $59.0 \pm 2.0$ | $59.6 \pm 1.3$ | $61.0 \pm 1.5$ | 59.1 |
| Meta-Baseline-SC (Chen et al., 2020b) | RN12 | $66.3 \pm 0.6$ | $73.3 \pm 1.3$ | $63.5 \pm 0.3$ | $63.9 \pm 0.8$ | 66.8 |
| Meta-Baseline-MoCo (Nie et al., 2021) | RN12 | $65.9 \pm 1.4$ | $72.2 \pm 0.8$ | $63.9 \pm 0.8$ | $64.7 \pm 0.3$ | 66.7 |
| WReN-Bongard (Barrett et al., 2018) | RN12 | $50.1 \pm 0.1$ | $50.9 \pm 0.5$ | $53.8 \pm 1.0$ | $54.3 \pm 0.6$ | 52.3 |
| CNN-Baseline (Nie et al., 2021) | RN12 | $51.9 \pm 0.5$ | $56.6 \pm 2.9$ | $53.6 \pm 2.0$ | $57.6 \pm 0.7$ | 54.9 |
| Meta-Baseline-PS (Nie et al., 2021) | RN12 | $68.2 \pm 0.3$ | $75.7 \pm 1.5$ | $67.4 \pm 0.3$ | $71.5 \pm 0.5$ | 70.7 |
| TPT (Shu et al., 2022) | CLIP | 52.5 | 65.9 | 58.6 | 56.3 | 58.3 |
| PredRNet (Yang et al., 2023) | Other | $\mathbf{74.6} \pm 0.3$ | $75.2 \pm 0.6$ | $\mathbf{71.1} \pm 1.5$ | $68.4 \pm 0.7$ | 72.3 |
| Prototype | RN12 | $68.9 \pm 0.5$ | $68.8 \pm 1.6$ | $64.7 \pm 0.8$ | $66.2 \pm 0.8$ | $67.1 \pm 0.7$ |
| Prototype + standardize | RN12 | $72.9 \pm 0.5$ | $80.8 \pm 0.5$ | $66.8 \pm 1.1$ | $71.3 \pm 1.4$ | $73.0 \pm 0.8$ |
| SVM | RN12 | $65.7 \pm 0.4$ | $75.3 \pm 1.3$ | $65.4 \pm 2.0$ | $70.4 \pm 1.6$ | $69.2 \pm 1.1$ |
| SVM + standardize | RN12 | $72.2 \pm 0.3$ | $83.8 \pm 0.9$ | $68.3 \pm 1.6$ | $71.8 \pm 1.2$ | $74.0 \pm 0.5$ |
| $k$-NN | RN12 | $62.1 \pm 0.7$ | $69.3 \pm 0.9$ | $62.8 \pm 0.6$ | $63.3 \pm 0.9$ | $64.4 \pm 0.4$ |
| $k$-NN + standardize | RN12 | $73.0 \pm 0.6$ | $\mathbf{84.9} \pm 0.8$ | $67.8 \pm 1.8$ | $71.0 \pm 1.3$ | $74.2 \pm 0.5$ |
| Prototype-Mimic | RN12 | $73.1 \pm 0.4$ | $80.9 \pm 0.6$ | $68.4 \pm 1.1$ | $72.0 \pm 0.9$ | $73.6 \pm 0.3$ |
| SVM-Mimic | RN12 | $73.3 \pm 0.3$ | $84.3 \pm 0.8$ | $69.4 \pm 0.8$ | $\mathbf{74.2} \pm 0.3$ | $\mathbf{75.3} \pm 0.1$ |
| Human Expert (Nie et al., 2021) | | $92.1 \pm 7.0$ | $99.3 \pm 1.9$ | $90.7 \pm 6.1$ | | 94.0 |
| Human Amateur (Nie et al., 2021) | | $88.0 \pm 7.6$ | $90.0 \pm 11.7$ | $71.0 \pm 9.6$ | | 83.0 |

## 4 Experiments

To train our models, we used the AdamW optimizer (Loshchilov & Hutter, 2017), a maximum learning rate of 5e−5, and a 1-cycle learning rate policy (Smith & Topin, 2019), increasing the learning rate linearly for the first 5% of the training steps. We ran all experiments on NVIDIA Titan RTX, Titan XP, GeForce, or A5000 GPUs. Each experiment used at most 1 GPU and 30GB of GPU memory.

### 4.1 Bongard-LOGO

**Dataset**   We evaluate on the Bongard-LOGO dataset (Nie et al., 2021), which consists of synthetic images containing geometric shapes similar to those in the original Bongard problems. We follow previous works (Nie et al., 2021) and use a ResNet-15 (He et al., 2016) trained from scratch. Each Bongard-LOGO problem consists of seven positive and seven negative examples. The first six positives and first six negatives are chosen to serve as supports, and the remaining two are used as queries.

We used the dataset splits defined by the dataset authors. The train, validation, and test sets consist of 9300, 900, and 1800 problems, respectively. The test set is further subdivided into four splits, each of which assesses various forms of out-of-domain generalization. In the "basic shape" split, each image contains two shapes that were seen separately in training but were not seen together. In the "free-form shape" split, each image contains a free-form shape that was not seen in training. The "combinatorial abstract shape" split uses seen shape attributes (e.g., "has four straight lines") in unseen combinations. The "novel abstract shape" split is similar, except one of the concepts was never seen in training. We used the validation set to select hyperparameters and report results on the test set splits.

Table 2: **Results on Bongard-HOI with a frozen CLIP encoder.** Error bars are obtained by evaluating three trained models over the test set (thousands of problems), for all methods except deterministic ones.

| Method | Unseen Act / Unseen Obj | Seen Act / Unseen Obj | Unseen Act / Seen Obj | Seen Act / Seen Obj | Average |
|---|---|---|---|---|---|
| HOITrans (Zou et al., 2021) | 62.87 | 64.38 | 63.10 | 59.50 | 62.46 |
| TPT (Shu et al., 2022) | 65.66 | 65.32 | 68.70 | 66.03 | 66.43 |
| BDC-Adapter (Zhang et al., 2023) | 67.82 | 67.67 | 69.15 | 68.36 | 68.25 |
| Prototype | 65.45 | 65.73 | 76.31 | 68.02 | 68.88 |
| Prototype + standardize | 68.11 | 68.92 | 77.44 | 69.97 | 71.11 |
| SVM | 68.05 | 67.36 | 74.89 | 67.91 | 69.55 |
| SVM + standardize | 69.48 | 69.50 | 75.37 | 69.68 | 71.01 |
| $k$-NN | 61.04 | 60.88 | 68.54 | 63.28 | 63.44 |
| $k$-NN + standardize | 66.72 | 67.08 | 75.65 | 68.04 | 69.37 |
| Prototype-Mimic | 68.88 $\pm$ 0.30 | 70.74 $\pm$ 0.44 | 76.66 $\pm$ 0.26 | **71.25** $\pm$ 0.29 | 71.88 $\pm$ 0.20 |
| SVM-Mimic | **69.59** $\pm$ 0.13 | **70.83** $\pm$ 0.27 | **78.13** $\pm$ 0.27 | 71.23 $\pm$ 0.07 | **72.45** $\pm$ 0.16 |
| Human (Jiang et al., 2023) | 87.21 | 90.01 | 93.61 | 94.85 | 91.42 |

**Training Details**  We train models for 500,000 iterations with a batch size of 2. We resize images to $512 \times 512$ pixels and apply random cropping and horizontal flipping to augment. We train all models using support dropout. Label noise did not improve validation results (likely because the ground truth is error-free), so we omit it.

**Results**  Table 1 shows results on Bongard-LOGO's four test splits. We observe that our simple baselines (Prototype, SVM, and $k$-NN) coupled with our contrastive encoder match or exceed most previous approaches. Incorporating standardization leads to a 5-10 point boost for each of them. The support-set Transformer models, Prototype-Mimic and SVM-Mimic, perform better than the non-Transformer variants. SVM-Mimic performs the best, with an accuracy of 75.3%. This is 3 points better than the best prior method PredRNet (Yang et al., 2023) and fewer than 8 points behind human amateurs. While PredRNet performs well on Bongard-LOGO, it is not straightforward to extend it to natural image tasks like Bongard-HOI, since it cannot leverage arbitrary vision backbones, and relies on training end-to-end from scratch. Results vary substantially between the four test splits as each split consists of a different distribution shift, requiring a different type of problem-solving of varying difficulty (humans find the combinatorial and novel abstract shape tests the hardest). Concurrently to the preparation of our work, Song and Yuan introduced three methods that achieve average accuracies in the 80% range on Bongard-LOGO (Song & Yuan, 2024a;b;c). We do not compare to these methods as they were evaluated using larger vision backbones (ResNet-18 and above). Our methods can be added to diverse baselines, as demonstrated in Table 1, so it is possible that Song and Yuan's methods would be improved by incorporating support-set standardization or Transformers. In contrast to us, Song and Yuan do not evaluate on natural image tasks such as Bongard-HOI.

## 4.2 Bongard-HOI

**Dataset**  Bongard-HOI (Jiang et al., 2023) structures Bongard problems around natural images of human-object interaction. The positive support set of each problem involves a human interaction with an object, and the negative support set contains different interactions with the same object. Following prior works (Shu et al., 2022; Zhang et al., 2023), we use CLIP with a ResNet-50 backbone as our image encoder. The Bongard-HOI dataset is structured similarly to the Bongard-LOGO dataset, with seven positives and seven negatives per problem. Since the dataset does not specify the support/query split for each problem, we arbitrarily select the final positive and negative images as queries.

The dataset authors defined train, validation, and test sets consisting of 23041, 17187, and 13941 problems, respectively. We manually created a clean subset of the original training and validation sets after observing that the publicly-available Bongard-HOI dataset is imbalanced and contains incorrect labels (details in Appendix). We publicly release this cleaned training dataset to aid future works. We did not clean the test set and instead use the dataset authors' test set without modifications. The test set consists of four splits: the "seen act/seen obj" split contains only human-object interactions where the actions and objects were both seen during training; "seen act/unseen obj" evaluates on problems with seen actions and unseen objects, and so on. We selected hyperparameters using the validation set and report results on the test sets. Due to incorrect labels in the test set, Bongard-HOI can be very challenging for even humans to solve (see qualitative examples in the Appendix). This motivates our use of label noise in training, which produces robustness to such incorrect labels.

**Training Details**  We use augmentations at training time, including horizontal flips, random grayscale, color jitter, and random rescaling and cropping (to $224 \times 224$), and apply support dropout and label noise. We train for 10,000 iterations with a batch size of 16 and weight decay of 0.01.

**Results**  Table 2 shows quantitative results on the four test splits of Bongard-HOI. (Many of the prior baselines in Table 1 have been applied on Bongard-HOI but attain very low performance (Jiang et al., 2023), so we exclude them for brevity.) We first note that our simple baselines (Prototype, SVM, and $k$-NN) approach or beat the state-of-the-art methods, TPT (Shu et al., 2022) and BDC-Adapter (Zhang et al., 2023). This is surprising, as none of these baselines uses training at any point, while TPT and BDC-Adapter are learning-based. For every baseline, standardization leads to further performance improvements. For example, Prototype baseline combined with standardization exceeds TPT by almost 3 points. Using a support-set Transformer leads to further improvements: SVM-Mimic exceeds its counterpart baseline, SVM + standardize, by almost 1.5 points.

### 4.3  Bongard-Classic

**Dataset**  Mikhail Bongard's original dataset (Foundalis, 2006; Yun et al., 2020) is a challenging dataset of roughly 200 problems with no training or validation set. Since there is no training set, we directly evaluate backbones and support-set Transformers pre-trained on Bongard-LOGO. We perform the same data pre-processing steps as done for Bongard-LOGO. Since the dataset has six positive and six negative examples for each problem, we select one positive and one negative example as queries. For each Bongard problem, we average results over every possible choice of positive and negative query and report accuracy, i.e., the percent of correct predictions. To our knowledge, prior works do not report results using accuracy as we define it. Grinberg et al. (2023) do not actually solve a query classification task as we do (and as done in Bongard-HOI and -LOGO) and instead search for a discrete rule which satisfies all supports, and they report the success rate of this search. Yun et al. (2020) solve a query classification task but use new manually created queries that they did not publish along with their work. We believe our accuracy metric is more reproducible.

**Results**  Table 3 shows the results of our approaches on Bongard-Classic. Due to the evaluation differences mentioned, we are unable to compare with other works but hope our results can serve as baselines for future work. Standardization leads to improvements in every case. Support-set Transformers do not lead to consistent improvements, which is expected as the Transformer has not been fine-tuned on Bongard-Classic and encounters a distribution shift. SVM-Mimic is our second-best method, attaining 60.84% accuracy.

### 4.4  Bongard-HOI Results with Alternative Vision Backbones

For completeness, we demonstrate the generality of our approaches by applying them on Bongard-HOI with vision backbones other than CLIP.

#### 4.4.1  Fine-Tuned CLIP Backbone

The state-of-the-art method for category-level few-shot learning tasks is PMF (Hu et al., 2022), and this technique involves fine-tuning the backbone encoder to produce better prototypes. This is complementary

Table 3: **Results on Bongard-Classic.** Error bars are obtained by testing three models on all problems.

| Method | Accuracy |
|---|---|
| Prototype | $57.52 \pm 0.36$ |
| Prototype + standardize | $57.82 \pm 0.72$ |
| SVM | $60.10 \pm 0.55$ |
| SVM + standardize | $\mathbf{61.27} \pm 0.12$ |
| $k$-NN | $56.80 \pm 0.44$ |
| $k$-NN + standardize | $58.10 \pm 0.33$ |
| Prototype-Mimic | $57.61 \pm 0.89$ |
| SVM-Mimic | $60.84 \pm 0.43$ |

Table 4: **Results on Bongard-HOI with CLIP fine-tuned via PMF**. For brevity, we average results over all test splits.

| Method | Accuracy |
|---|---|
| PMF (Hu et al., 2022) | $74.44 \pm 0.22$ |
| Prototype + standardize + PMF | $75.83 \pm 0.20$ |
| SVM + standardize + PMF | $75.42 \pm 0.17$ |
| Prototype-Mimic + PMF | $75.60 \pm 0.14$ |
| SVM-Mimic + PMF | $\mathbf{76.41} \pm 0.14$ |

Table 5: **Results on Bongard-HOI with a frozen DINO encoder.**

| Method | Accuracy |
|---|---|
| Prototype | 72.51 |
| Prototype + standardize | 73.88 |
| SVM | 73.29 |
| SVM + standardize | 73.54 |
| $k$-NN | 68.34 |
| $k$-NN + standardize | 71.76 |
| Prototype-Mimic | $72.66 \pm 0.30$ |
| SVM-Mimic | $\mathbf{74.08} \pm 0.10$ |

to our approach. Table 4 shows results from combining our methods with PMF, evaluated on Bongard-HOI. PMF has not previously been applied to Bongard problems, and it cannot be fairly compared with the methods in Table 2 due to the fine-tuned rather than frozen encoder. We combine support-set standardization, support-set Transformers, and our baselines with PMF by first fine-tuning the CLIP encoder with the PMF objective, freezing the weights, and then plugging the fine-tuned encoder directly into each method. PMF performs very well, exceeding the results obtained with frozen CLIP. Support-set standardization and Transformers lead to further improvements. One failure case is Prototype-Mimic; note that the PMF training objective already optimizes the prototypes, so it is possible that the forward pass by the Transformer is not necessary. SVM-Mimic combined with PMF performs best. PMF training details are in the Appendix.

### 4.4.2 DINO Backbone

We next use a DINO ViT-base backbone (Caron et al., 2021) instead of a CLIP-based one. Table 5 contains results with a DINO backbone on Bongard-HOI. Support-set standardization leads to consistent performance improvements for each of the three baselines. SVM-Mimic achieves a further performance boost of 0.54 points compared to SVM + standardize. One exception to the performance improvements is Prototype-Mimic, which fails to improve upon Prototype + standardize.

### 4.5 Robustness

To further demonstrate the success of our methods, we measure the robustness of SVM-Mimic. First, we report robustness to the number of support samples provided as input. Figure 3 shows the results of SVM-Mimic, a version of SVM-Mimic trained without support dropout or label noise (SVM-Mimic w/o Reg), and SVM baseline + standardization (SVM + Std) on the test splits of Bongard-HOI. SVM-Mimic

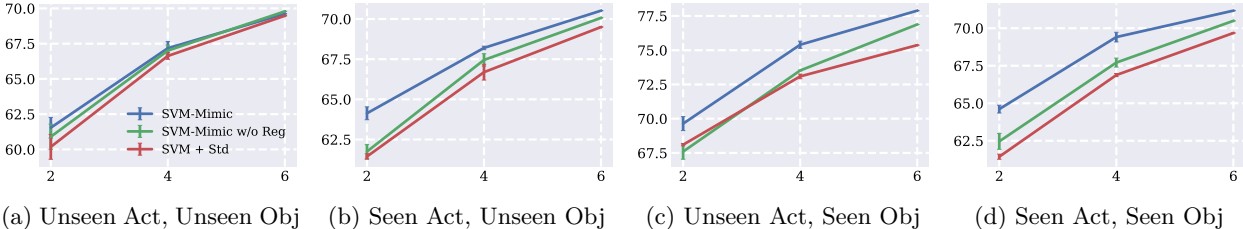

(a) Unseen Act, Unseen Obj     (b) Seen Act, Unseen Obj     (c) Unseen Act, Seen Obj     (d) Seen Act, Seen Obj

Figure 3: **Robustness to number of supports in Bongard-HOI.** The x-axis measures the number of supports of each class seen, with 6 being the default setting, and the y-axis measures accuracy. Means and standard deviations are across three runs, where randomness is with respect to the subset of supports chosen.

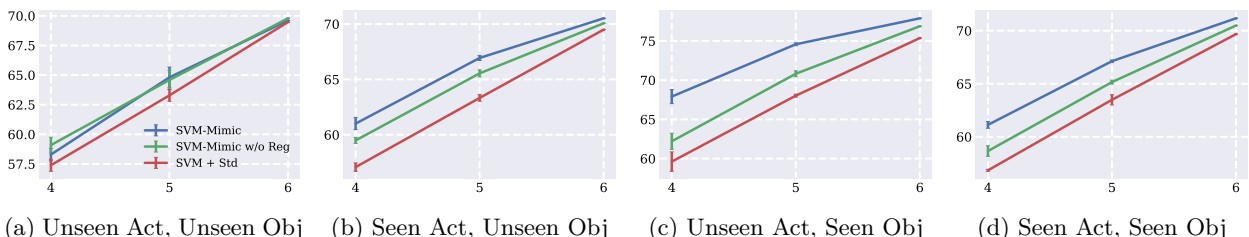

(a) Unseen Act, Unseen Obj     (b) Seen Act, Unseen Obj     (c) Unseen Act, Seen Obj     (d) Seen Act, Seen Obj

Figure 4: **Robustness to label noise in Bongard-HOI.** The x-axis measures the number of supports of each class that have **not** been noised. Other details are the same as in Figure 3.

attains consistently higher accuracy than the other approaches for different numbers of supports. Figure 4 demonstrates the robustness of SVM-Mimic to noise in support labels, where noise is created by flipping the labels for random choices of positive and negative supports. In almost all cases, SVM-Mimic is clearly more robust than SVM-Mimic w/o Reg and SVM + Std. These results demonstrate the benefits of learning a Transformer to produce rules rather than non-parametrically fitting an SVM to each Bongard problem.

## 4.6 Comparison to Autoregressive Vision-Language Models

Large autoregressive vision-language foundation models such as GPT-4V (Achiam et al., 2023) and GPT-4o (OpenAI, 2024) have demonstrated strong reasoning capabilities, which raises the question: how well do our simple methods compare to these state-of-the-art foundation models on Bongard problems? GPT-4o is expensive to evaluate, so we evaluated it on the dataset where we have the most baselines, Bongard-LOGO. GPT-4o performs poorly compared to SVM-Mimic, as shown in Table 6. This may be partly due to the significant distribution shift between Bongard-LOGO (synthetic abstract images) and GPT-4o's training dataset (likely Internet images), so it is possible that GPT-4o would perform better on Bongard-HOI. In concurrent work by Zhao et al. (2023), a new vision-language foundation model "MMICL" was proposed, and evaluated on a subset of Bongard-HOI. We compare this value to SVM-Mimic + PMF in Table 7, and find that SVM-Mimic performs more than 2 points better than MMICL. MMICL, which uses InstructBLIP (Dai et al., 2023) as its backbone, likely does not suffer the same distribution shift on Bongard-HOI as GPT-4o does on Bongard-LOGO, so it is notable that our simple methods still outperform it.

**Implementation details**    We used the "gpt-4o" model in the OpenAI API. To have the model classify a query image, we included a prompt with all support images and labels, the unlabeled query, and an explanation of the classification objective. We experimented with multiple variations of the prompt on a subset of the Bongard-LOGO validation set, including (i) instructing the model to "think step-by-step" prior to guessing (Kojima et al., 2022) and (ii) few-shot prompting (Brown et al., 2020), where the "shots" are the support examples. The results in Table 6 use few-shot prompting, which gave the best performance on the validation subset. The final prompt is in the Appendix.

Table 6: **Ours vs. GPT-4o on Bongard-LOGO.** Results are averaged over the four test splits.

| Method | Accuracy |
|--------|----------|
| GPT-4o (OpenAI, 2024) | 65.0 |
| SVM-Mimic | **75.3** $\pm$ 0.1 |

Table 7: **Ours vs. MMICL on Bongard-HOI.** The MMICL result is evaluated by the MMICL authors and only uses a subset of the test set, while the SVM-Mimic result uses the entire test set.

| Method | Accuracy |
|--------|----------|
| MMICL (Zhao et al., 2023) | 74.20 |
| SVM-Mimic + PMF | **76.41** $\pm$ 0.14 |

Table 8: **Effect of different forms of normalization in Bongard-HOI.** Results are averaged over the four test splits. Note that Prototype and $k$-NN use $l^2$ normalization by default.

| Method | Accuracy |
|--------|----------|
| Prototype | 68.88 |
| Prototype + train-set standardize | 70.06 |
| Prototype + support-set standardize | **71.11** |
| SVM | 69.55 |
| SVM + $l^2$ normalization | 67.94 |
| SVM + train-set standardize | **71.19** |
| SVM + support-set standardize | 71.01 |
| $k$-NN | 63.44 |
| $k$-NN + train-set standardize | 65.86 |
| $k$-NN + support-set standardize | **69.37** |

### 4.7 Ablation Study: Forms of Normalization

Can support-set standardization be replaced with other forms of centering and scaling? Table 8 compares the performance of different forms of normalization on Bongard-HOI. "Support-set standardize" is our main approach; "train-set standardize" computes mean and variance statistics using the entire training set of CLIP embeddings; $l^2$ normalization normalizes each embedding independently. Importantly, neither train-set standardization nor $l^2$ normalization adapts features using *task*-level context (instead focusing on the dataset-level and image-level). Train-set standardization leads to a 0.1 point gain over support-set standardization when using SVMs, but it performs 1 point worse with Prototypes and over 3 points worse with $k$-NN, suggesting that support-set standardization is a safe choice. We also find that $l^2$ normalization harms performance on the SVM, despite the fact that this is a design step included by default in Prototype and $k$-NN methods for computing cosine similarity.

## 5 Discussion and Conclusion

In this work, we point out that simple inductive biases to incorporate support-set context can lead to significant performance improvements on Bongard problems. We demonstrate that support-set standardization, coupled with simple baselines, in many cases achieves state-of-the-art performance on Bongard-LOGO and Bongard-HOI. Second, we explore *learning* to attend to support-set context with a Transformer and show that this leads to further improvements in accuracy. After controlling for vision backbone architecture, our best model, SVM-Mimic, sets a new state-of-the-art on Bongard-LOGO and Bongard-HOI.

One limitation of our approach is sensitivity to the choice of vision backbone. The Transformer learns to be robust to errors or ambiguities in the individual representations, but with stronger representations this may not be needed. This explains why Prototype-Mimic fails to improve upon Prototype + standardize for the DINO and PMF encoders (Tables 4 and 5), which both attain stronger baseline performance than CLIP does. Another limitation is that the Transformer is learned and may not generalize to new distributions upon which no training set exists. This explains why the Transformer does not improve upon baselines in Bongard-Classic, where we did no training. Finally, we note that our Bongard-LOGO results appear to have benefited greatly from our contrastive learning stage, and perhaps other methods could be improved by incorporating this feature-learning technique.

Overall, our experiments attest to the importance of support-set context for Bongard problems. We hope that our results, our models, and our code, will facilitate future work in this domain.

**Broader impact**  Our work describes improvements to make neural networks better at a certain visual IQ task. As AI systems approach human accuracy on this and other tasks, AI researchers will need to carefully consider the risks associated with having machines take over work traditionally performed by humans, such as the economic and emotional impact on the individuals that would normally do the work. As the limitations of AI systems are lifted, particularly in symbolic reasoning tasks such as those considered here, there is eventual risk of superhuman general-purpose task-solving, for which safeguards will need to be set in place. These ethical concerns are not limited to Bongard problems in particular.

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

## A    Appendix

This appendix is intended to aid reproduction of our results and provide additional insight into performance of our methods. We first elaborate on model architectures and training details. Second, we give an overview of dataset licenses and cleaning procedures. Third, we demonstrate additional results. We finally provide extensive qualitative results.

## B    Experimental Details

**Transformer Architecture**    For both Bongard-LOGO and Bongard-HOI, we use a Transformer encoder with a depth of six, eight heads per layer, and 64 dimensions per head. We set the Transformer token and MLP dimensions to match the dimensionality of vision backbone outputs. For Bongard-HOI with a CLIP ResNet-50 backbone, this is 1024. For Bongard-LOGO with a ResNet-15 backbone, this is 128. For task outputs (prototypes and rules), we perform layer normalization on the Transformer's output for the `cls` token(s) and transform them with a linear layer.

**PMF Training Details**    To train PMF, we use a maximum learning rate of $5e-7$ and a batch size of 4 and train for 40,000 iterations. We use neither support dropout nor label noise. In order to train Prototype-Mimic using the PMF backbone, we observed that it was necessary to train for 40,000 iterations instead of the usual 10,000 iterations. We did not need to make any changes to the SVM-Mimic training procedure when using a PMF backbone.

**DINO Training Details**    To obtain results using a DINO backbone, we use the same training procedure as for CLIP. We set the token and MLP dimensions at all layers of the Transformer to 768 to match DINO's output dimensionality. We found it necessary to train SVM-Mimic for 20,000 iterations and Prototype-Mimic for 30,000 iterations.

**Support Dropout Details**    When using train-time support dropout, we always retain at least two supports from each class and at most all supports. We sample the same number of supports for each class.

**Label Noise Details**    When using train-time label noise, we only flip the labels of one positive and one negative support. We found it beneficial to apply label noise to only 25% of the training batches at random.

**GPT-4o Prompting Details**    Our best-performing GPT-4o prompting tactic is few-shot prompting. The following is an example prompt on a Bongard-LOGO problem; note that the last shown image is the query and that we randomized the order of the supports:

- **System:** You are about to solve a type of visual reasoning problem called a Bongard problem. You will be shown 6 "positive" images followed by 6 "negative" images. The positive images each display some visual concept, and the negative images don't display the visual concept. After, you will be shown a single "query" image. Your goal is to deduce the concept in play and classify the query image as positive or negative, depending on whether it demonstrates the concept. As the examples demonstrate, output only a single word as your answer: "positive" or "negative".

- **User:** 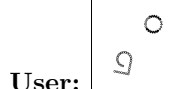

- **Assistant:** positive

- (other positive support examples)

- **User:** 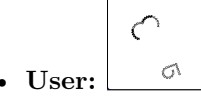

Table 9: **Original Bongard-HOI training dataset vs. our cleaned dataset**. Error bars are obtained by evaluating three trained models over the entire test set (thousands of problems).

| Method (Both w/o Dropout) | Unseen Act / Unseen Obj | Seen Act / Unseen Obj | Unseen Act / Seen Obj | Seen Act / Seen Obj | Avg |
|---|---|---|---|---|---|
| SVM + standardize | 69.48 | 69.50 | 75.37 | 69.68 | 71.01 |
| SVM-Mimic, Full | **69.88** ± 0.46 | 70.48 ± 0.20 | 77.24 ± 0.12 | 70.73 ± 0.32 | 72.08 ± 0.25 |
| SVM-Mimic, Clean | 69.59 ± 0.13 | **70.83** ± 0.27 | **78.13** ± 0.27 | **71.23** ± 0.07 | **72.45** ± 0.16 |

- **Assistant:** negative

- (other negative support examples)

- **User:** 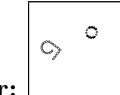

## C   Additional Dataset Details

**Dataset Licenses**   The datasets used in our experiments are all publicly available. The Bongard-LOGO dataset uses an MIT License. The Bongard-HOI dataset is open-sourced for research purposes. The dataset of problems from Mikhail Bongard (called Bongard-Classic in the paper) does not have a known license.

**Cleaned Bongard-HOI Dataset**   We observed that the Bongard-HOI dataset contains many incorrect labels and suffers from class imbalance. To mitigate this, we created a cleaned version of the train set and all validation sets. All SVM-Mimic results in our work were trained and validated on these cleaned sets but evaluated on the original Bongard-HOI test sets for fairest comparison with prior works. We manually curated these cleaned datasets. To do so, we first shuffled each Bongard problem's positive and negative sets, and randomly selected one image in each set to serve as the positive and negative queries. For easy manual inspection, we cropped every image to the human-object-interaction of interest using annotations provided in the dataset, and resized the resulting image to a square. We then discarded all Bongard problems that did not meet all of the following criteria: (i) the object should be very clearly present in both query images, (ii) the positive query should demonstrate the interaction indicated in the label, and (iii) the negative query should not demonstrate the interaction. For faster curation, we did not inspect the support images. To ensure a balanced dataset, we sampled at most 100 Bongard problems for each unique human-object interaction in the train set, and 20 for each interaction in the validation sets. Our cleaned train set contained 2,196 problems, while the original train set contained 23,041 problems.

Table 9 contains results for SVM-Mimic models trained on the original and the cleaned train sets. These results demonstrate a very small but nearly consistent improvement from using the cleaned train set (despite the smaller size). The improvement justifies our use of the cleaned train set, but the small magnitude difference suggests that this cleaning is not critical. Both models outperform SVM + standardize.

## D   Qualitative Examples

In this section, we provide several qualitative examples on each dataset. Each example is a single Bongard problem with a positive and a negative query. We additionally report a score for each query using the margin between the query's embedding and SVM-Mimic's predicted hyperplane. Higher magnitude scores indicate greater distance between the query embedding and the hyperplane, which reflects greater confidence. To be classified correctly, positive queries should have a positive score and negative queries should have a negative score.

We compute these scores as $\frac{w \cdot f + b}{|w|}$, where $w$ and $b$ are the coefficients and intercept of the hyperplane and $f$ is the feature-space embedding of the query.

### D.1   Bongard-HOI

We include examples on both the unseen action/unseen object test set as well as the unseen action/seen object test set.

### D.2   Bongard-LOGO

We include examples on all four test splits of Bongard-LOGO.

### D.3   Bongard-Classic

We additionally include a few examples on the challenging Bongard-Classic dataset.

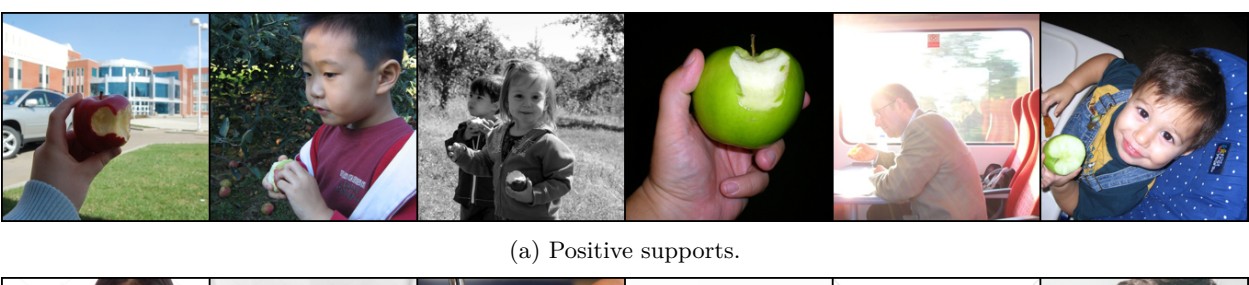

(a) Positive supports.

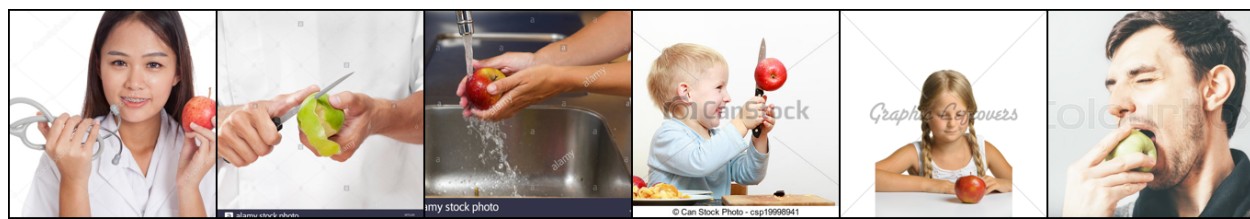

(b) Negative supports.

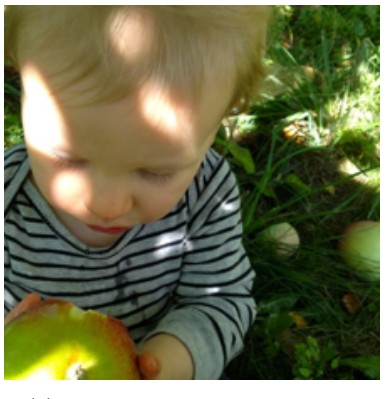

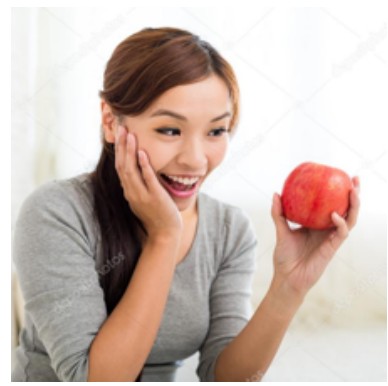

(c) Positive query. Score: 10.7.

(d) Negative query. Score: -10.3.

Figure 5: **Bongard-HOI unseen act/unseen obj: Correct guess for both queries**. The concept is "hold and about to eat apple."

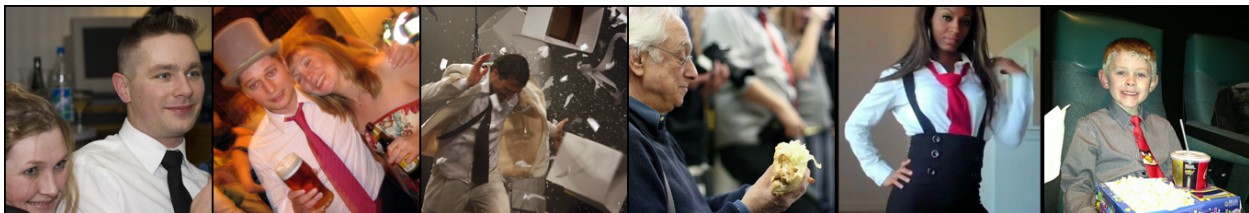

(a) Positive supports.

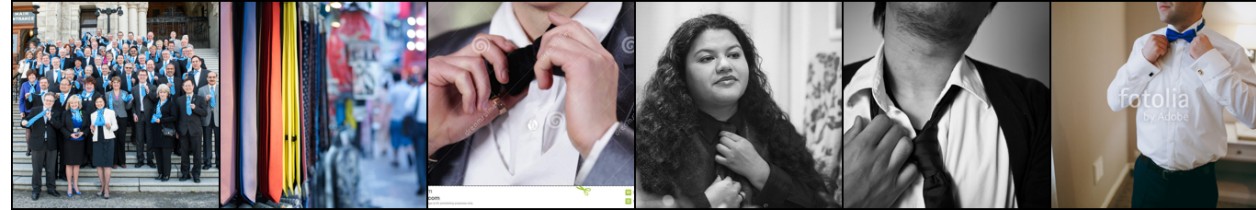

(b) Negative supports.

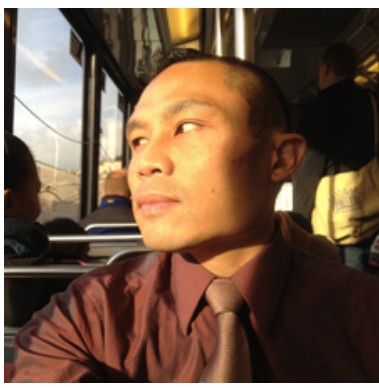 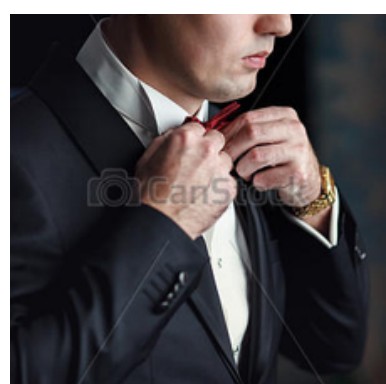

(c) Positive query. Score: 3.7.      (d) Negative query. Score: -6.8.

Figure 6: **Bongard-HOI unseen act/unseen obj: Correct guess for both queries**. The concept is "wear tie."

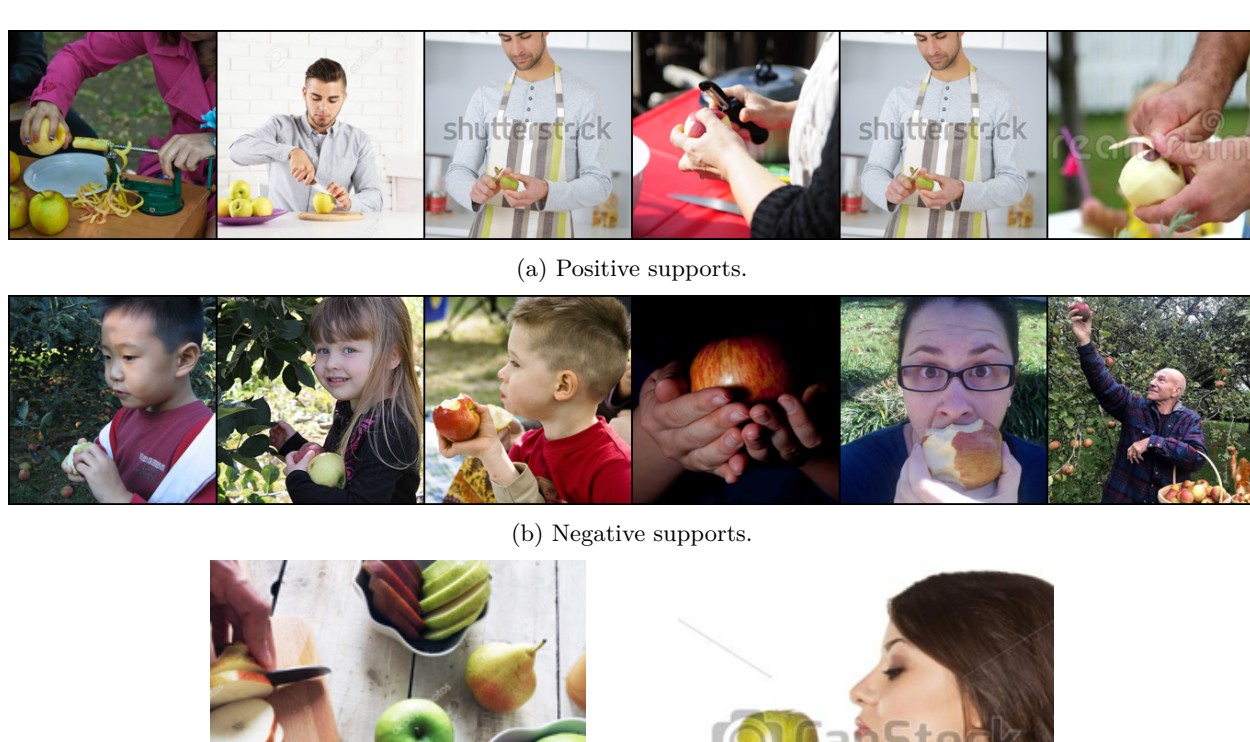

(a) Positive supports.

(b) Negative supports.

(c) Positive query. Score: -1.2.

(d) Negative query. Score: 9.8.

Figure 7: **Bongard-HOI unseen act/unseen obj: Incorrect guess for both queries**. The concept is "peel or cut apple."

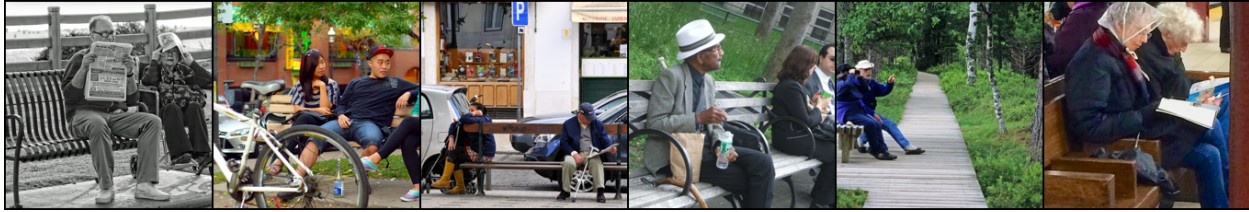

(a) Positive supports.

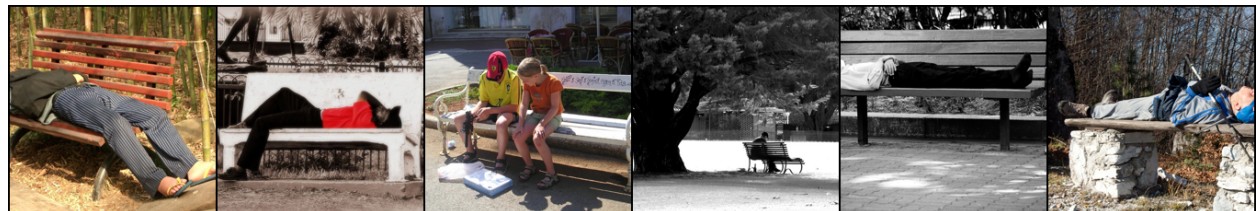

(b) Negative supports.

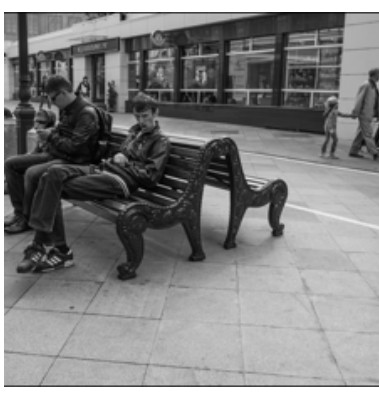

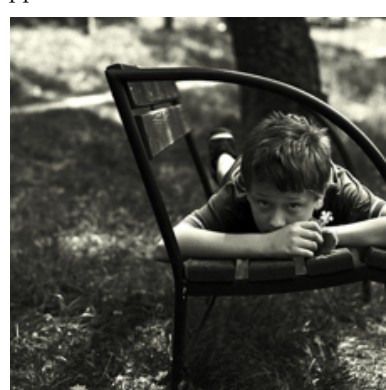

(c) Positive query. Score: -0.5.

(d) Negative query. Score: -1.9.

Figure 8: **Bongard-HOI unseen act/unseen obj: Incorrect for positive, correct for negative**. The concept is "sit on with multiple person bench."

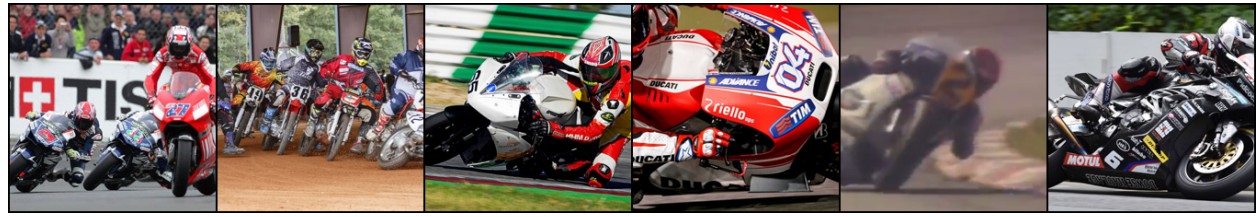

(a) Positive supports.

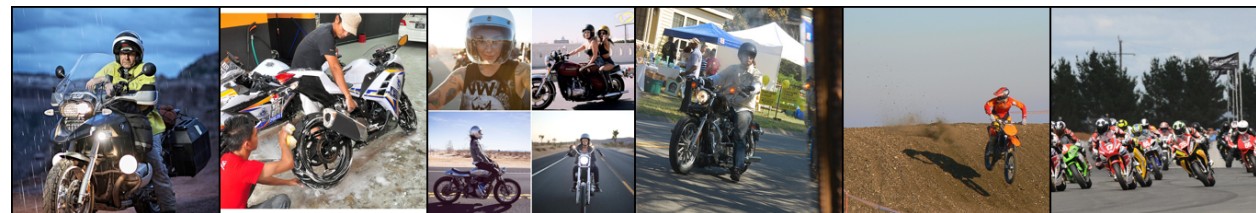

(b) Negative supports.

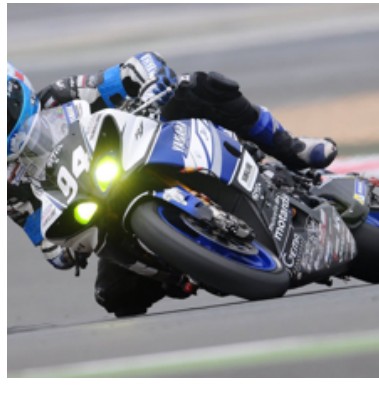

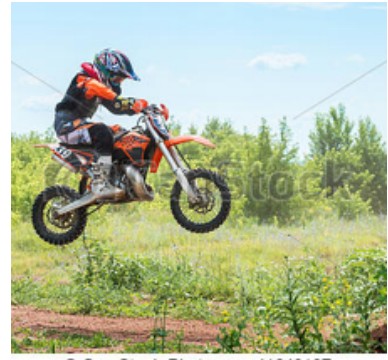

(c) Positive query. Score: 9.0.

(d) Negative query. Score: -4.2.

Figure 9: **Bongard-HOI unseen act/seen obj: Correct guess for both queries**. The concept is "turn motorcycle."

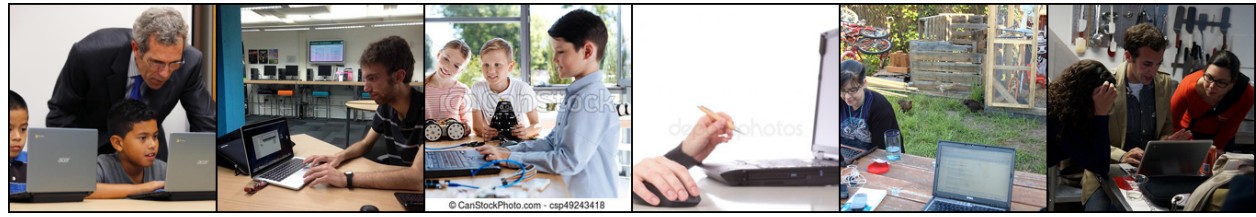

(a) Positive supports.

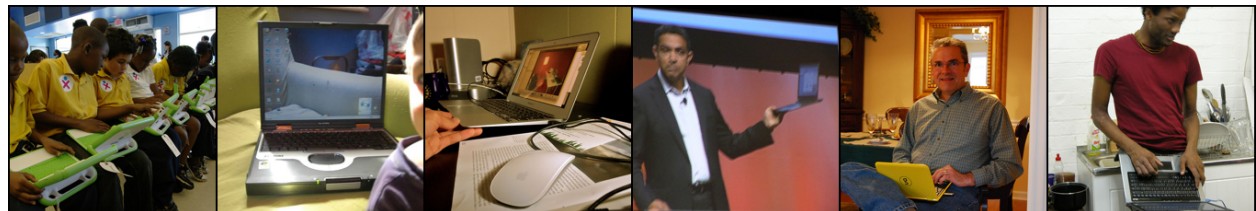

(b) Negative supports.

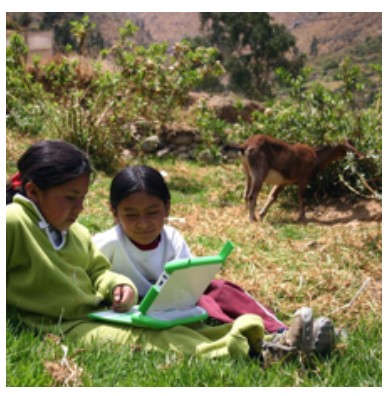

(c) Positive query. Score: 0.4.

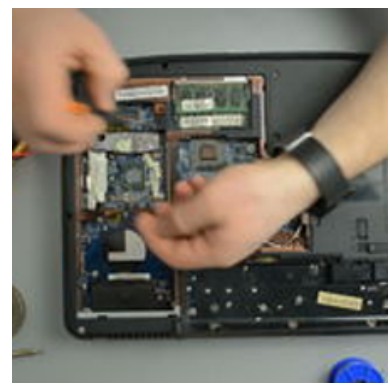

(d) Negative query. Score: -2.1.

Figure 10: **Bongard-HOI unseen act/seen obj: Correct guess for both queries**. The concept is "read laptop."

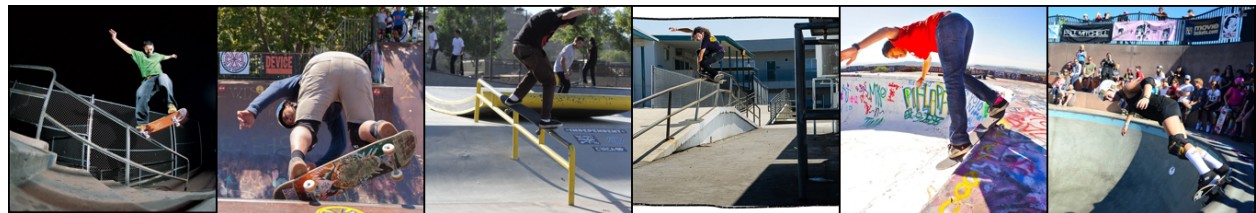

(a) Positive supports.

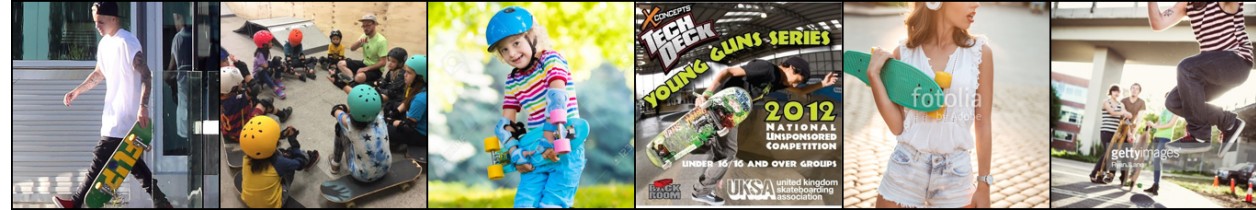

(b) Negative supports.

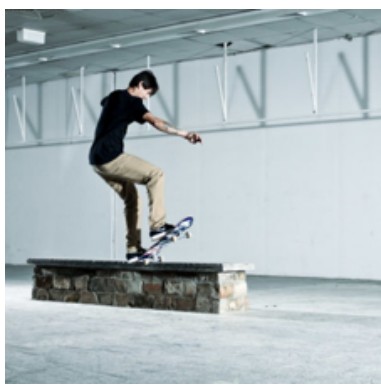
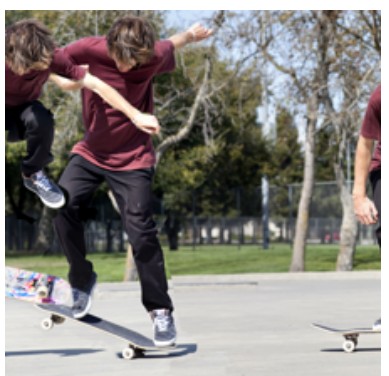

(c) Positive query. Score: 3.0.  (d) Negative query. Score: 5.6.

Figure 11: **Bongard-HOI unseen act/seen obj: Correct for positive, incorrect for negative**. The concept is "grind skateboard."

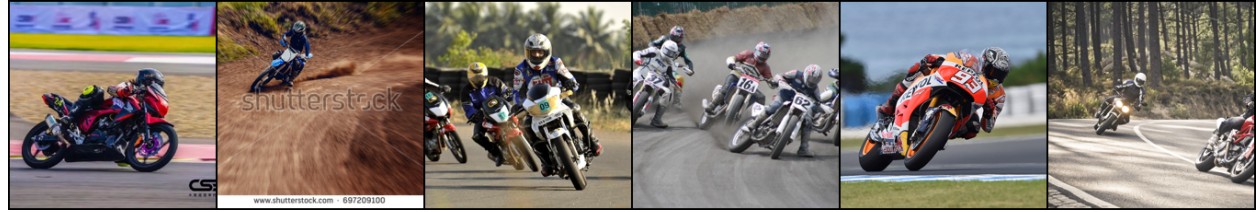

(a) Positive supports.

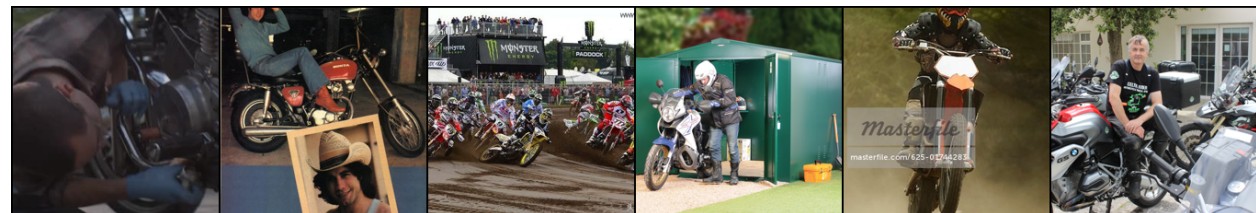

(b) Negative supports.

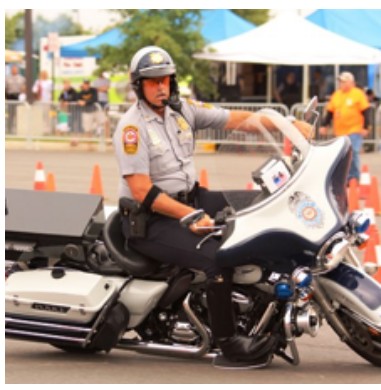

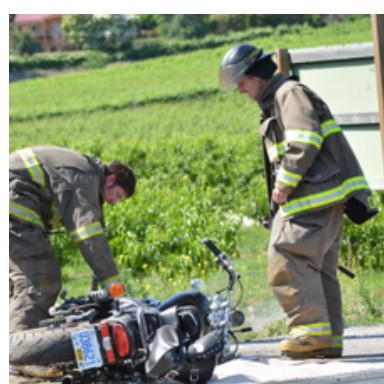

(c) Positive query. Score: -8.7.

(d) Negative query. Score: -7.1.

Figure 12: **Bongard-HOI unseen act/seen obj: Incorrect for positive, correct for negative**. The concept is "turn motorcycle."

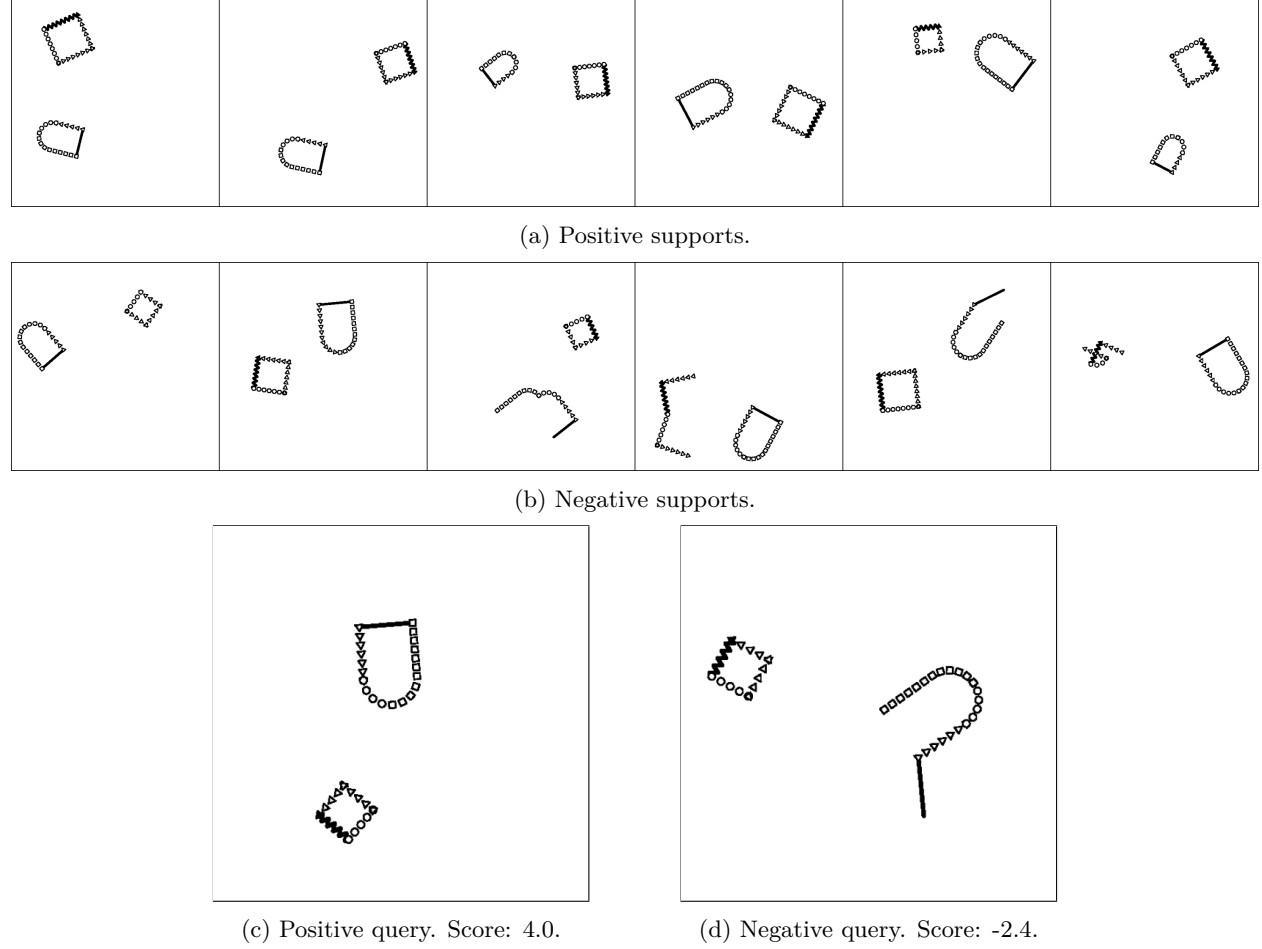

(a) Positive supports.

(b) Negative supports.

(c) Positive query. Score: 4.0.

(d) Negative query. Score: -2.4.

Figure 13: **Bongard-Logo Free-Form Shape: Correct guess for both queries**.

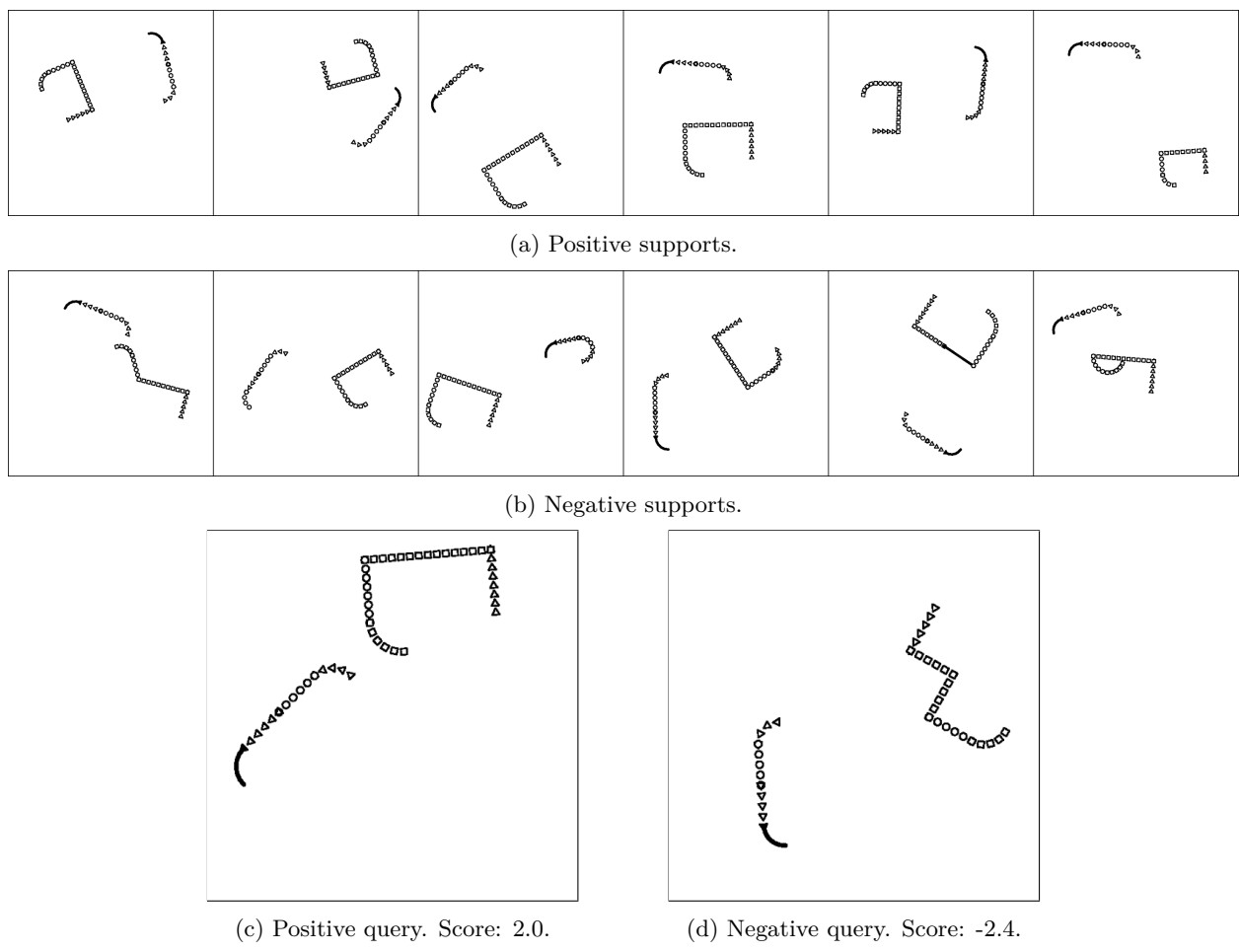

(a) Positive supports.

(b) Negative supports.

(c) Positive query. Score: 2.0.  (d) Negative query. Score: -2.4.

Figure 14: **Bongard-Logo Free-Form Shape: Correct guess for both queries**.

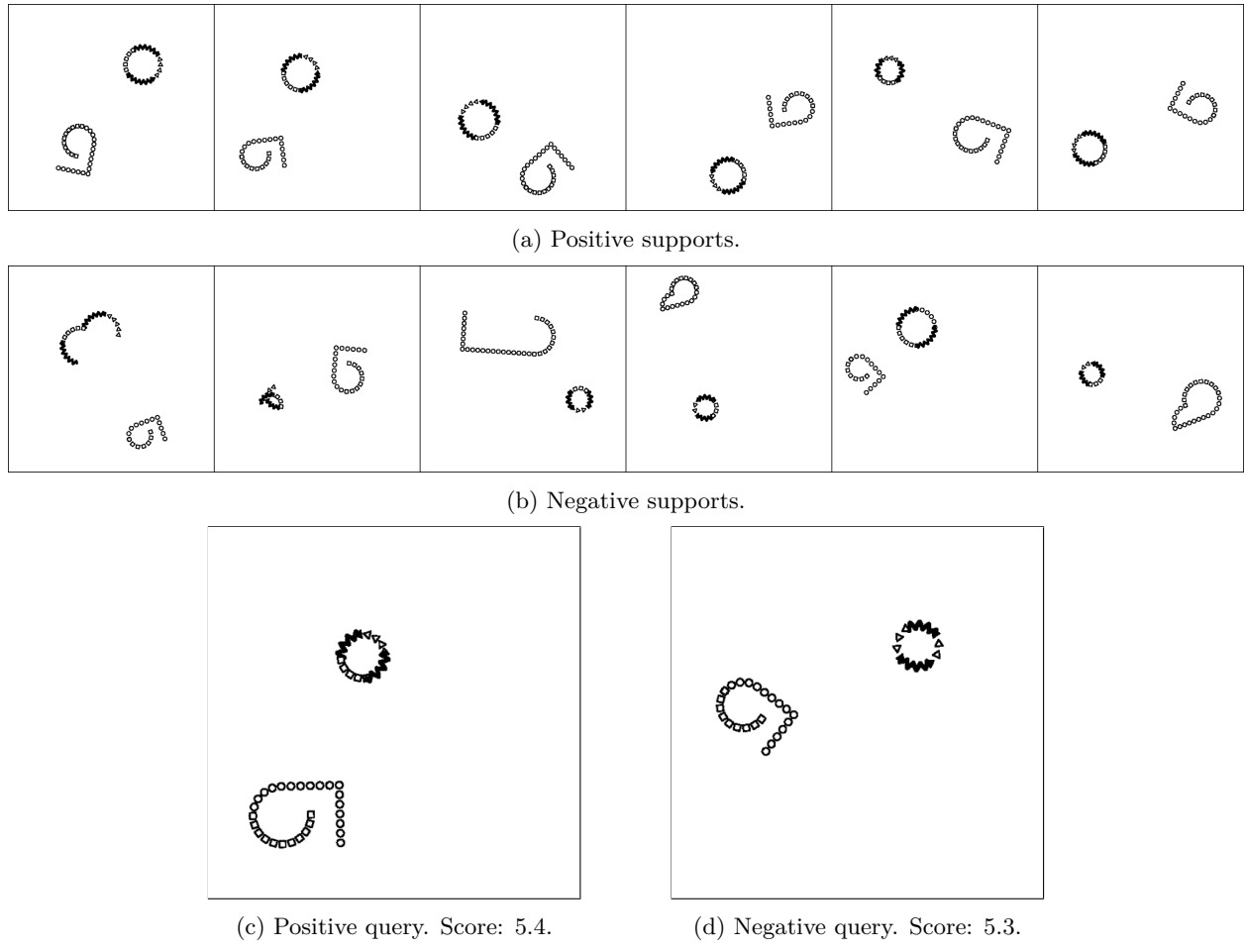

(a) Positive supports.

(b) Negative supports.

(c) Positive query. Score: 5.4.

(d) Negative query. Score: 5.3.

Figure 15: **Bongard-Logo Free-Form Shape: Correct for positive, incorrect for negative**.

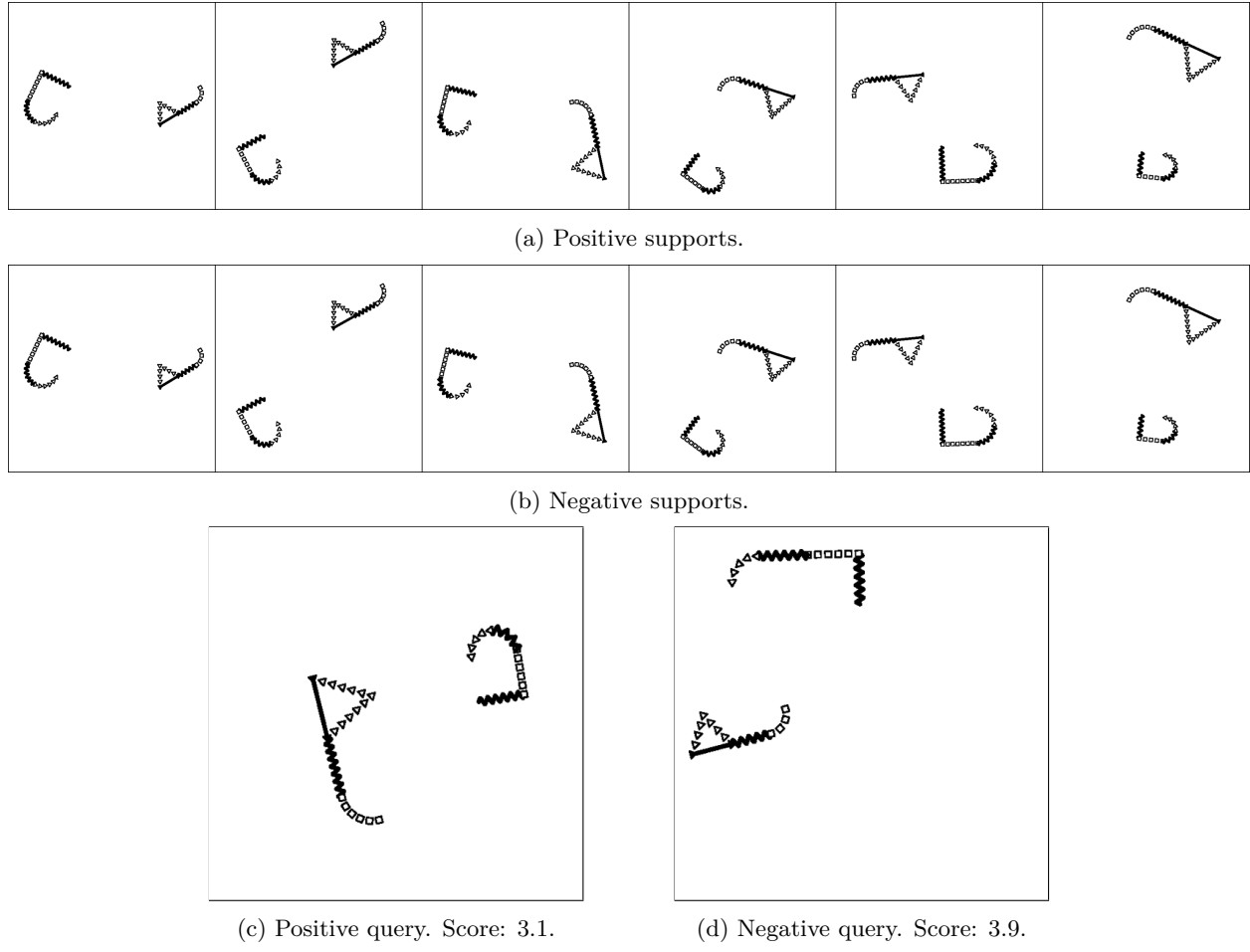

(a) Positive supports.

(b) Negative supports.

(c) Positive query. Score: 3.1.

(d) Negative query. Score: 3.9.

Figure 16: **Bongard-Logo Free-Form Shape: Correct for positive, incorrect for negative**.

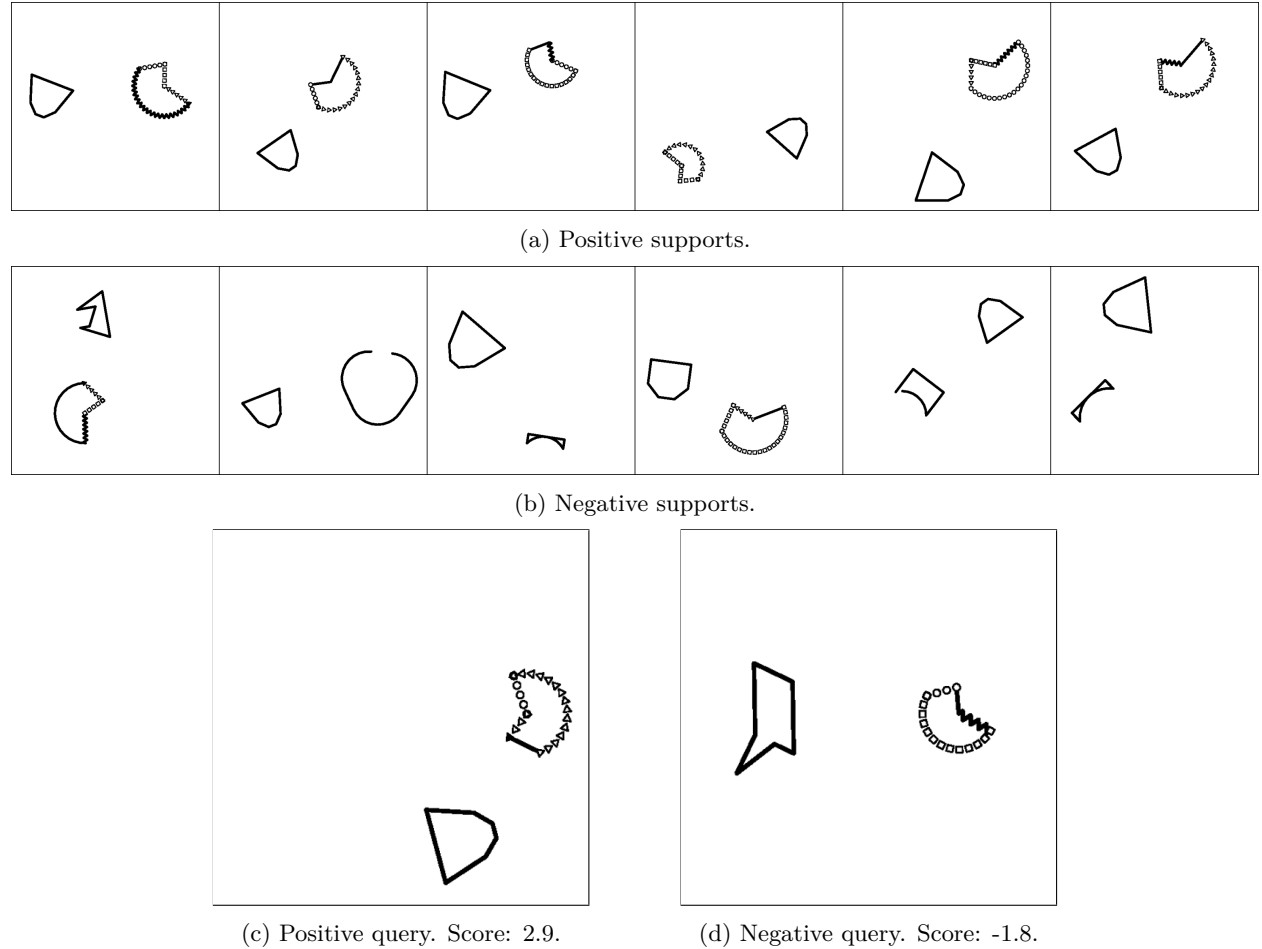

(a) Positive supports.

(b) Negative supports.

(c) Positive query. Score: 2.9.          (d) Negative query. Score: -1.8.

Figure 17: **Bongard-Logo Basic Shape: Correct guess for both queries**.

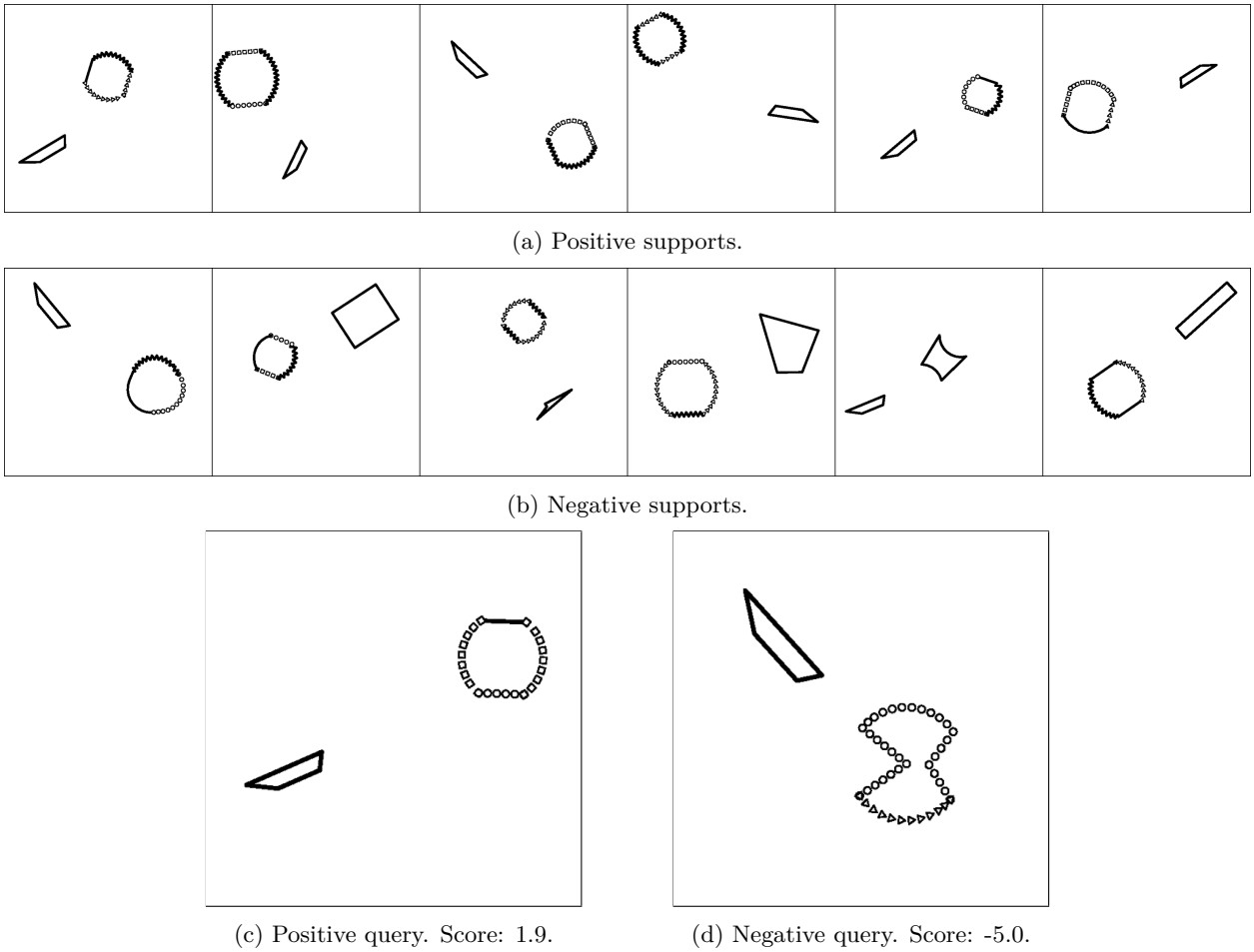

(a) Positive supports.

(b) Negative supports.

(c) Positive query. Score: 1.9.

(d) Negative query. Score: -5.0.

Figure 18: **Bongard-Logo Basic Shape: Correct guess for both queries**.

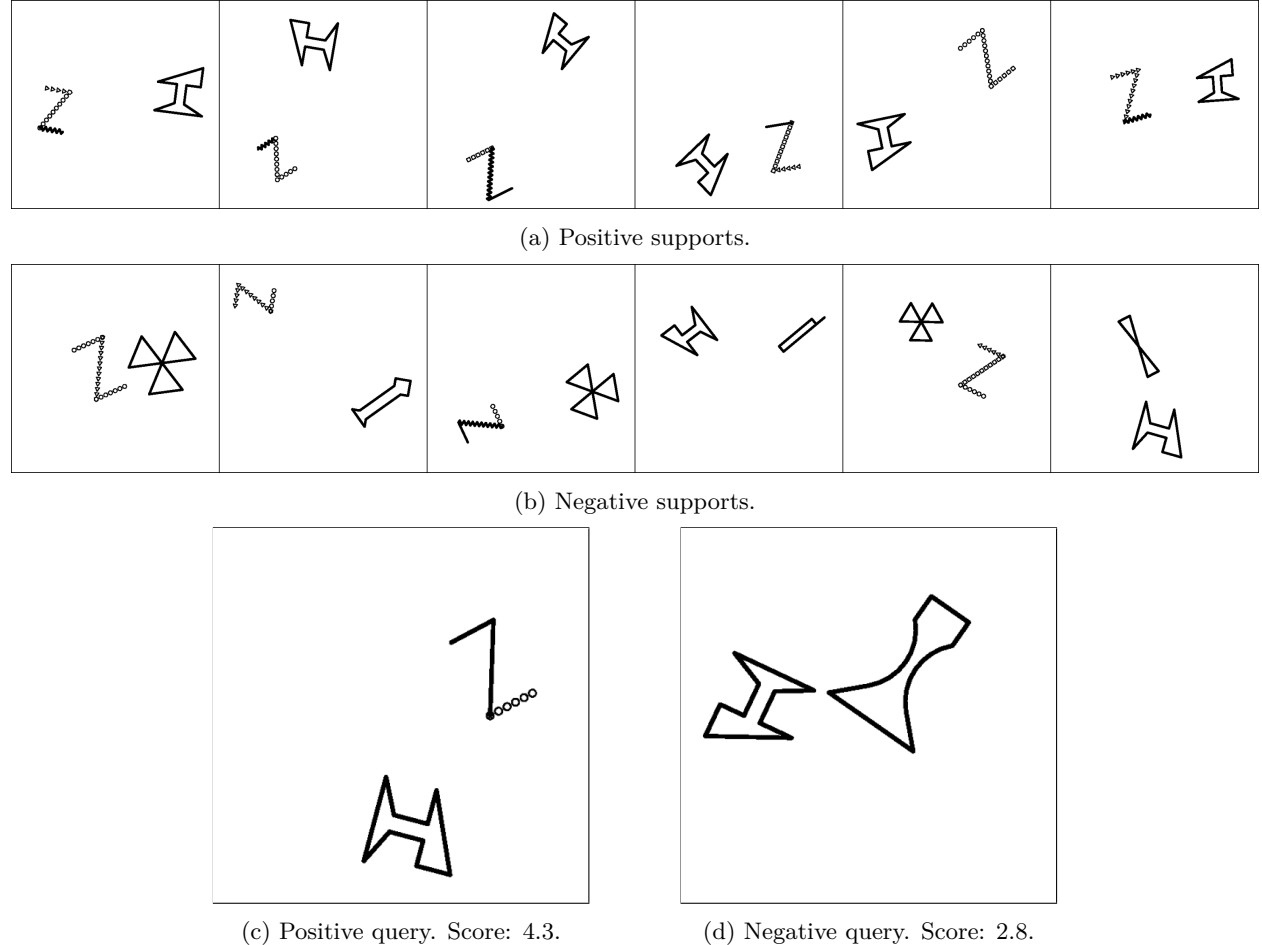

(a) Positive supports.

(b) Negative supports.

(c) Positive query. Score: 4.3.

(d) Negative query. Score: 2.8.

Figure 19: **Bongard-Logo Basic Shape: Correct for positive, incorrect for negative**.

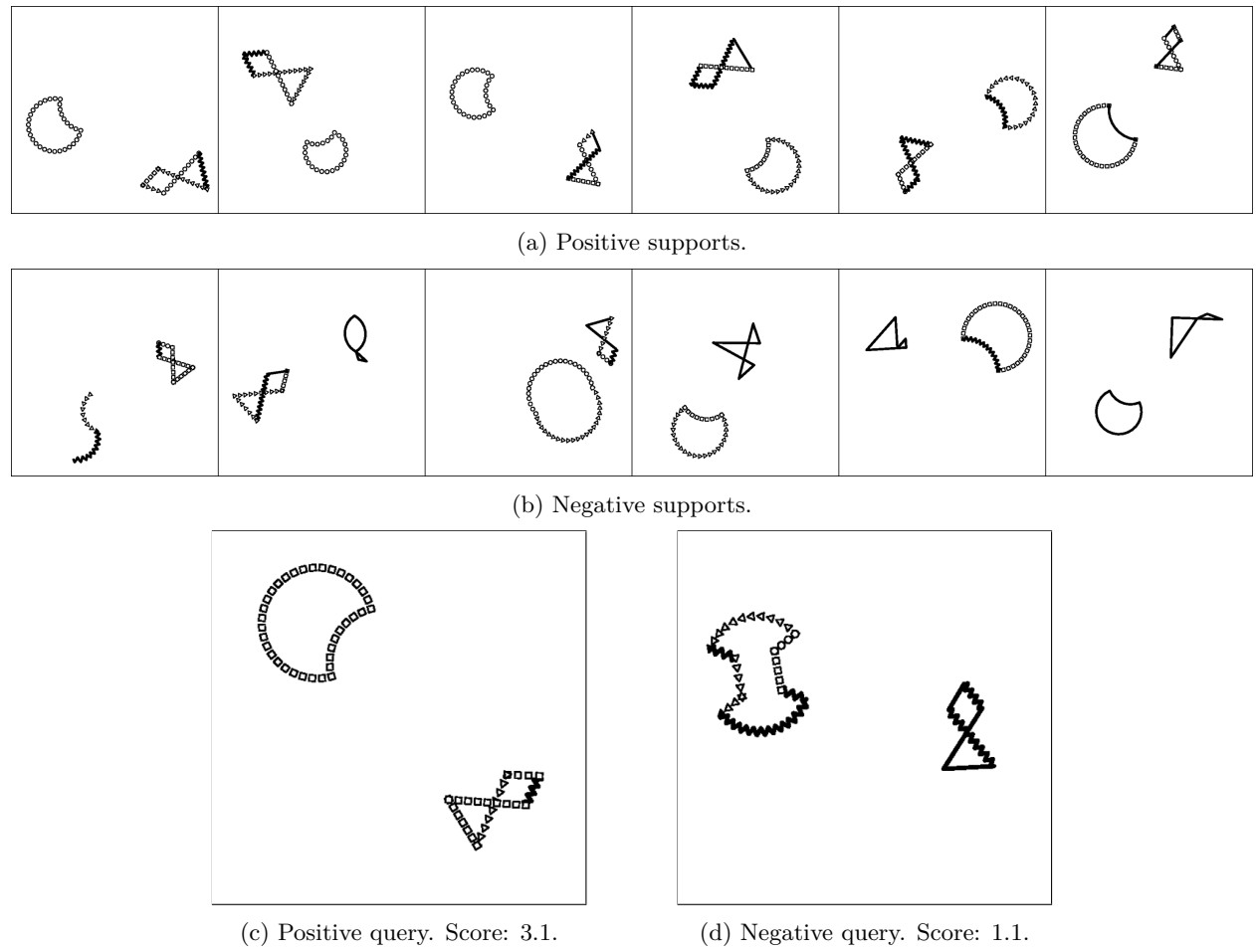

(a) Positive supports.

(b) Negative supports.

(c) Positive query. Score: 3.1.

(d) Negative query. Score: 1.1.

Figure 20: **Bongard-Logo Basic Shape: Correct for positive, incorrect for negative**.

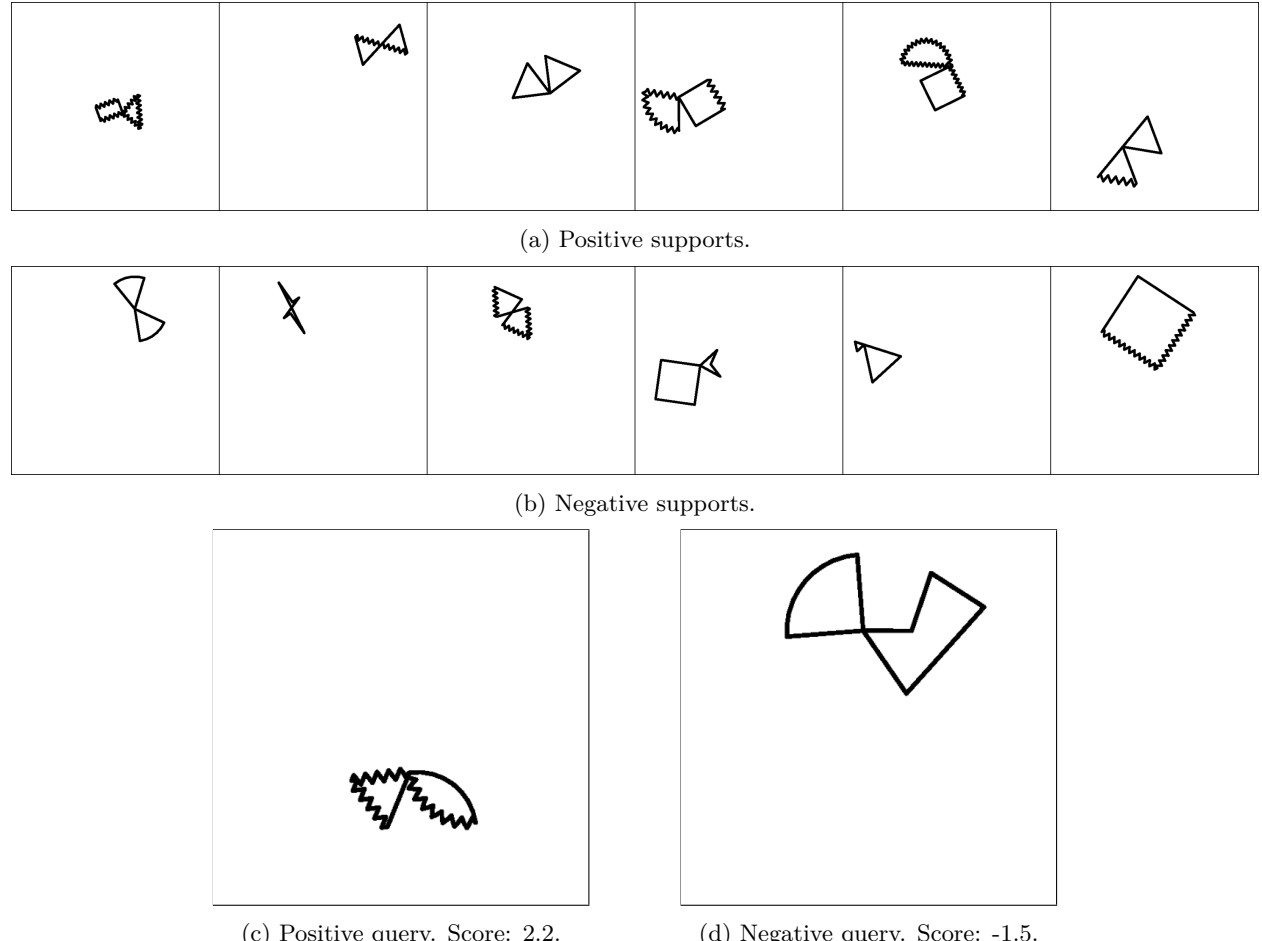

(a) Positive supports.

(b) Negative supports.

(c) Positive query. Score: 2.2.    (d) Negative query. Score: -1.5.

Figure 21: **Bongard-Logo Combinatorial Abstract Shape: Correct guess for both queries**.

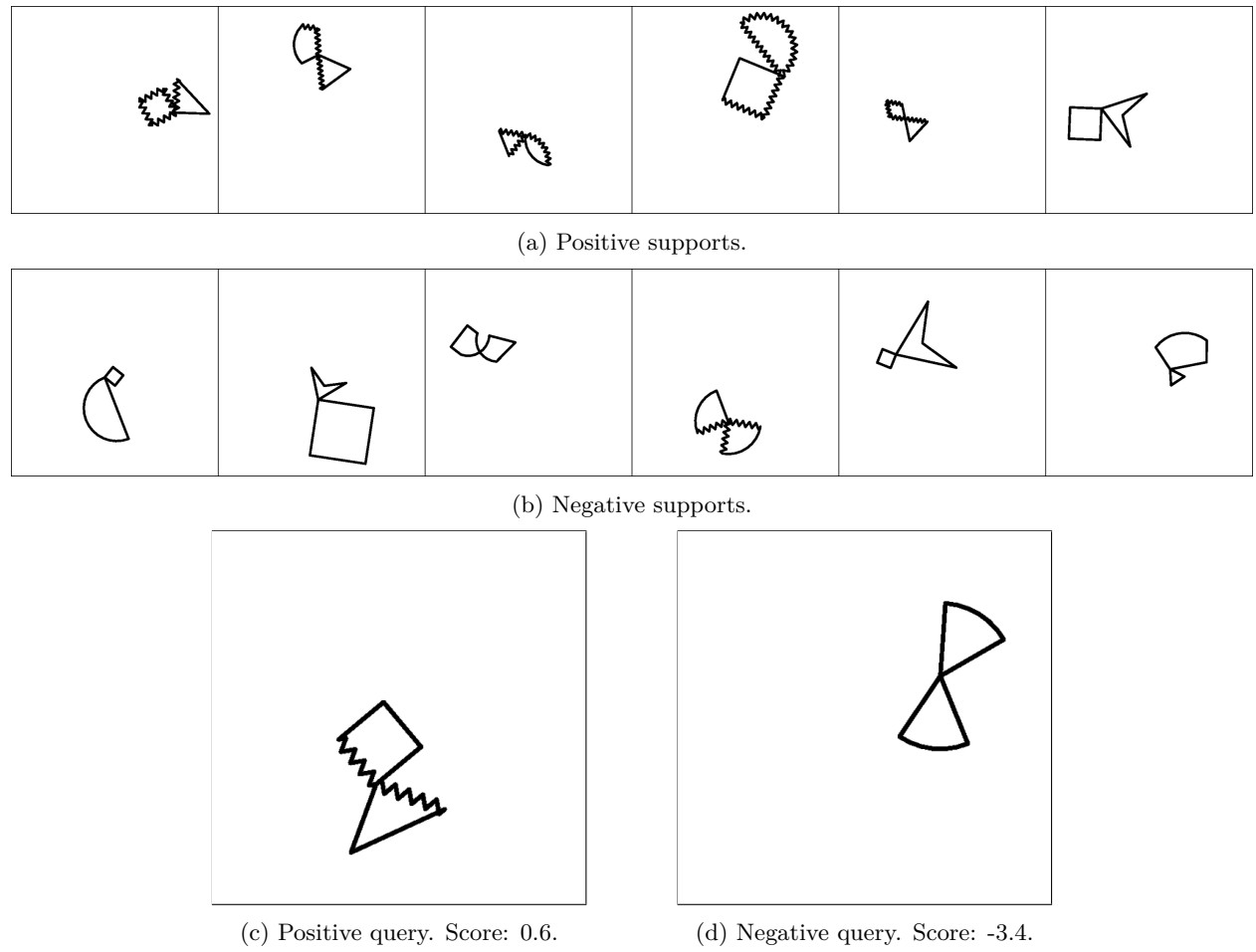

(a) Positive supports.

(b) Negative supports.

(c) Positive query. Score: 0.6.

(d) Negative query. Score: -3.4.

Figure 22: **Bongard-Logo Combinatorial Abstract Shape: Correct guess for both queries**.

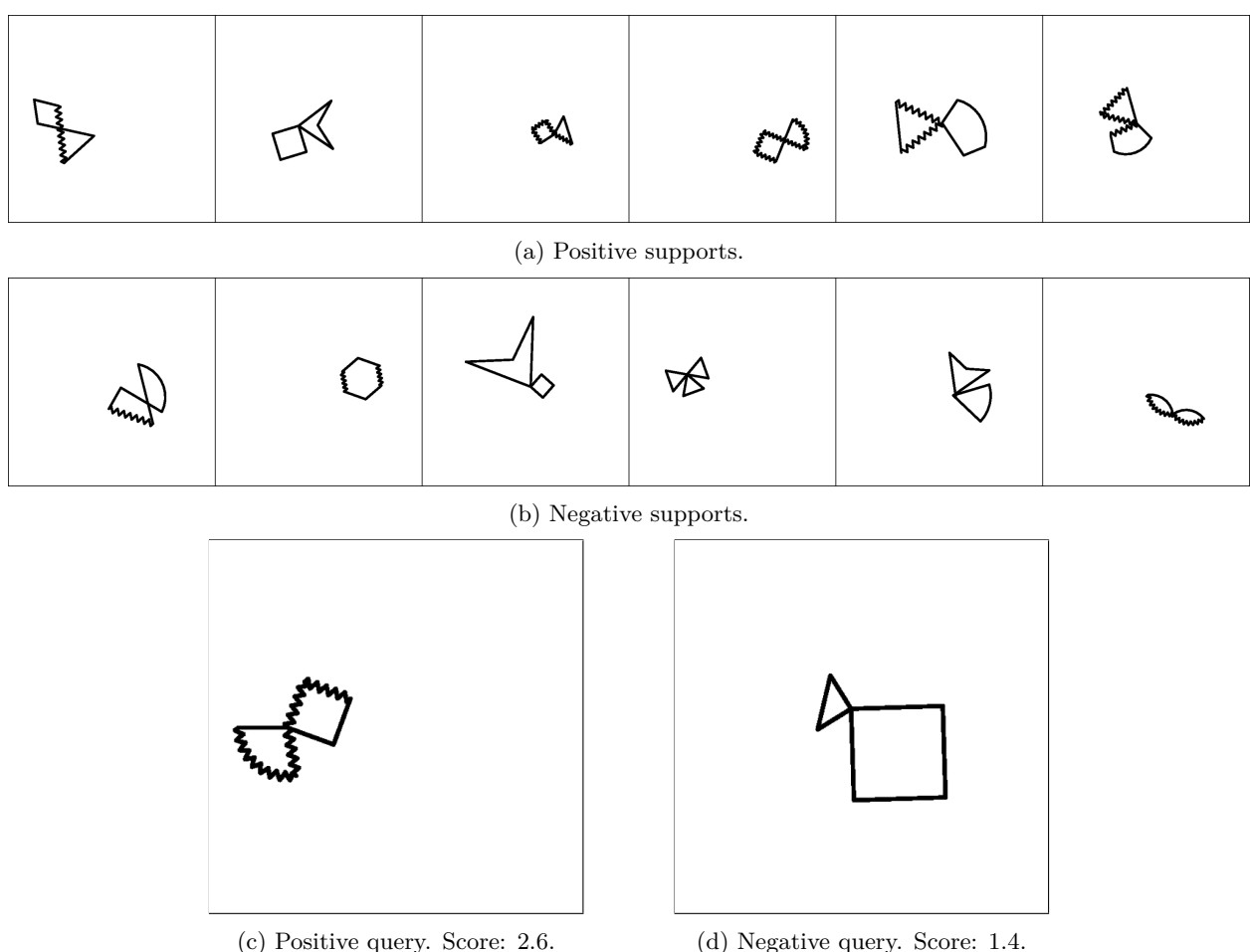

(a) Positive supports.

(b) Negative supports.

(c) Positive query. Score: 2.6.    (d) Negative query. Score: 1.4.

Figure 23: **Bongard-Logo Combinatorial Abstract Shape: Correct for positive, incorrect for negative**.

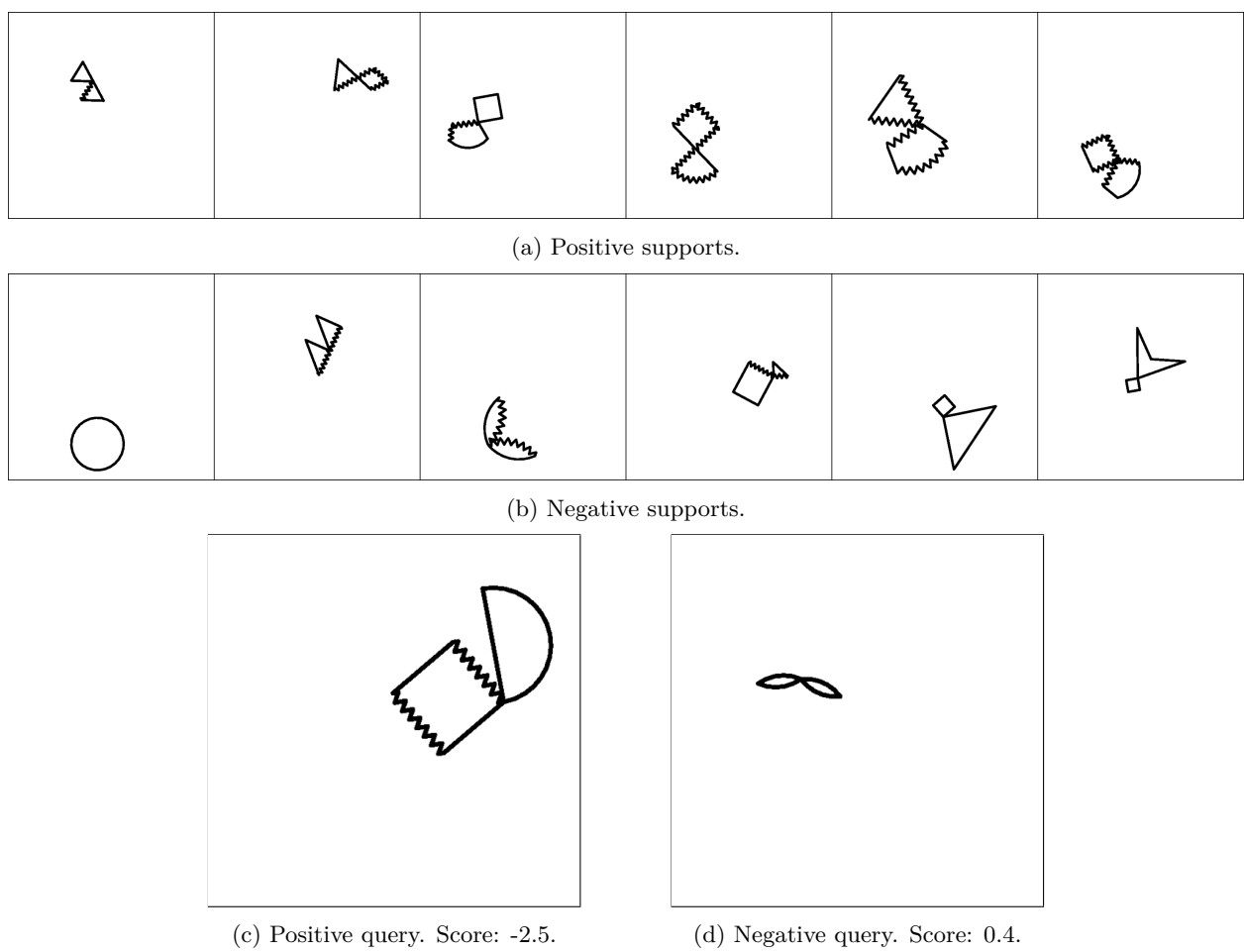

(a) Positive supports.

(b) Negative supports.

(c) Positive query. Score: -2.5.     (d) Negative query. Score: 0.4.

Figure 24: **Bongard-Logo Combinatorial Abstract Shape: Incorrect guess for both queries**.

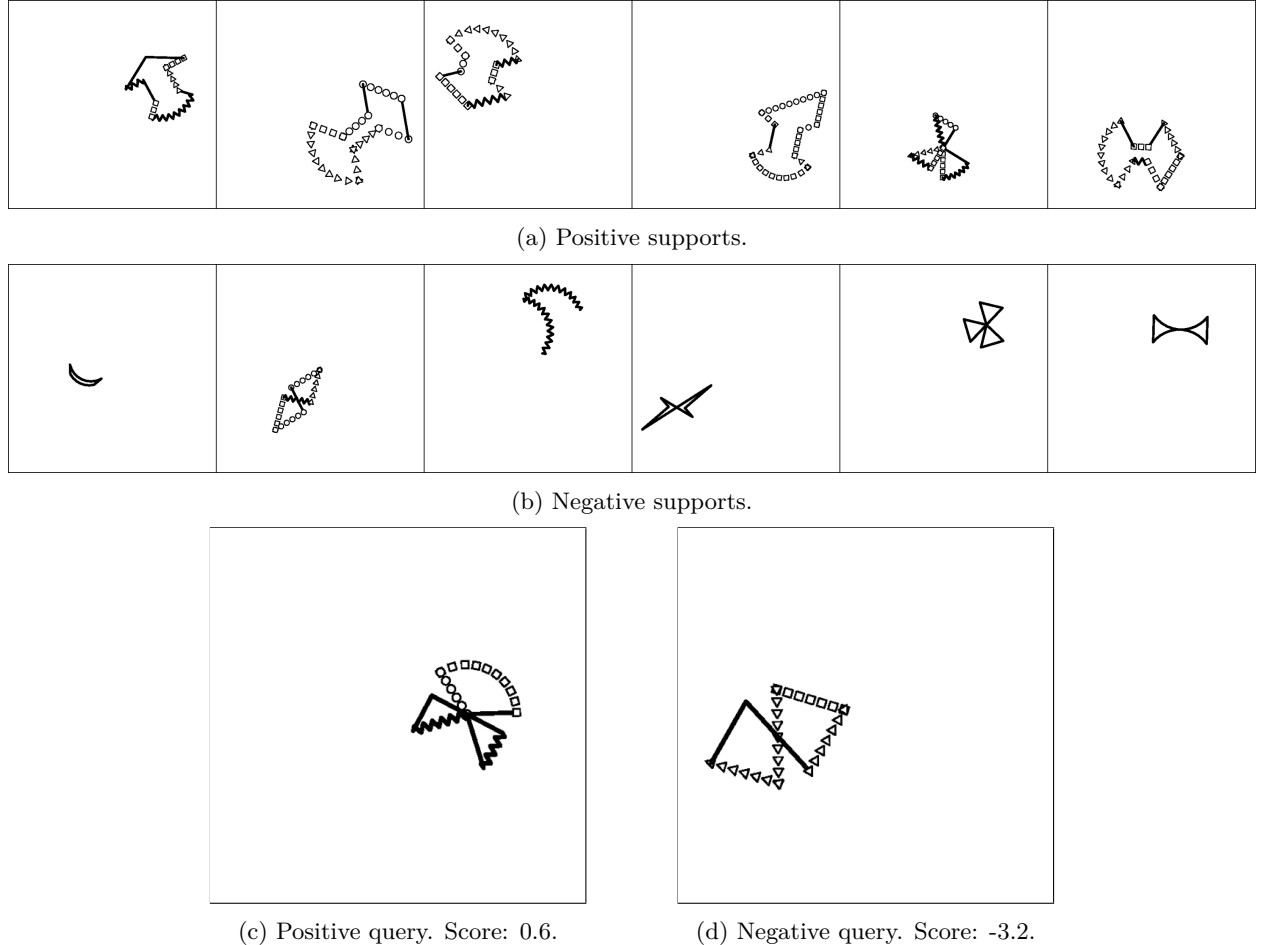

(a) Positive supports.

(b) Negative supports.

(c) Positive query. Score: 0.6.  (d) Negative query. Score: -3.2.

Figure 25: **Bongard-Logo Novel Abstract Shape: Correct guess for both queries**.

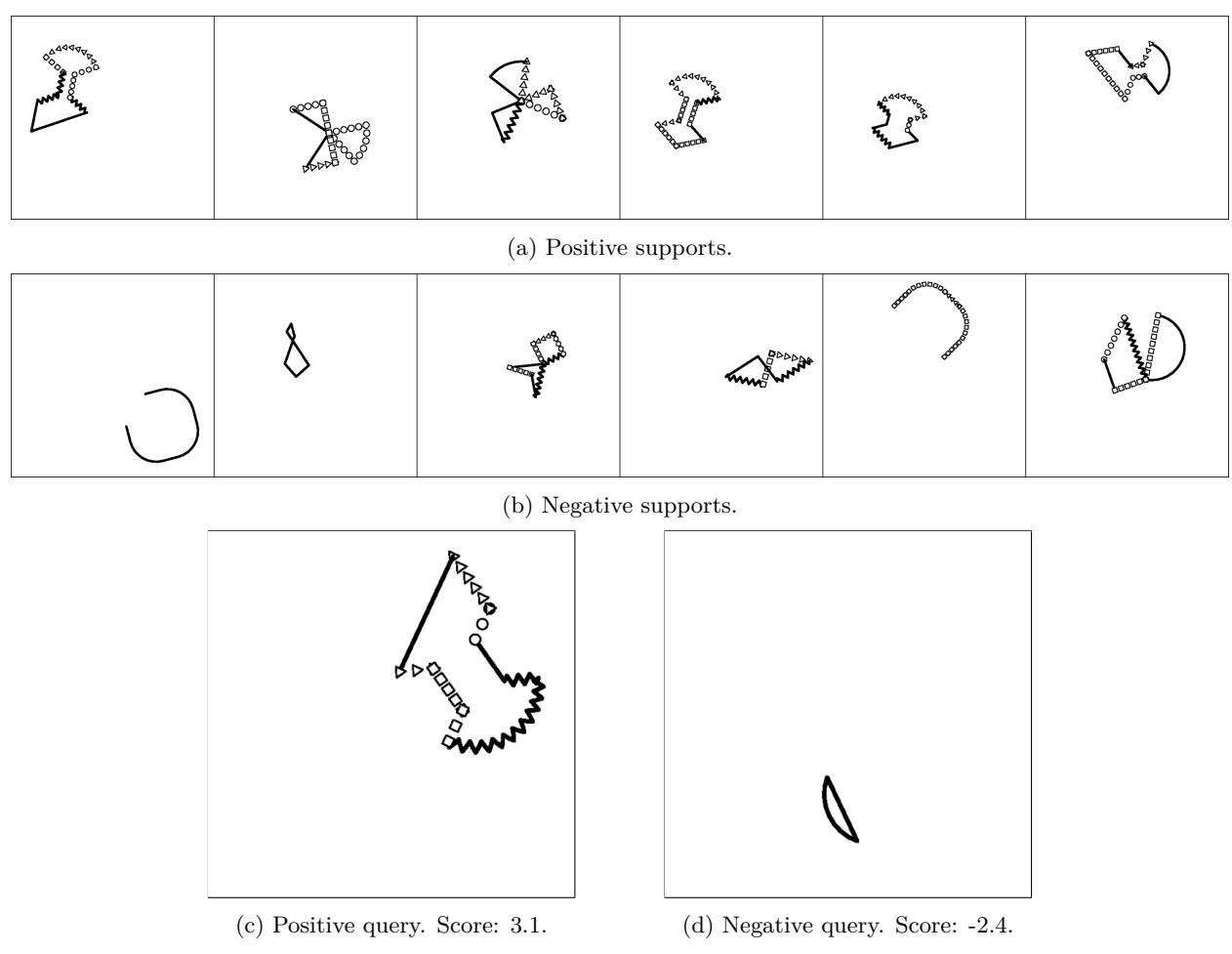

(a) Positive supports.

(b) Negative supports.

(c) Positive query. Score: 3.1.

(d) Negative query. Score: -2.4.

Figure 26: **Bongard-Logo Novel Abstract Shape: Correct guess for both queries**.

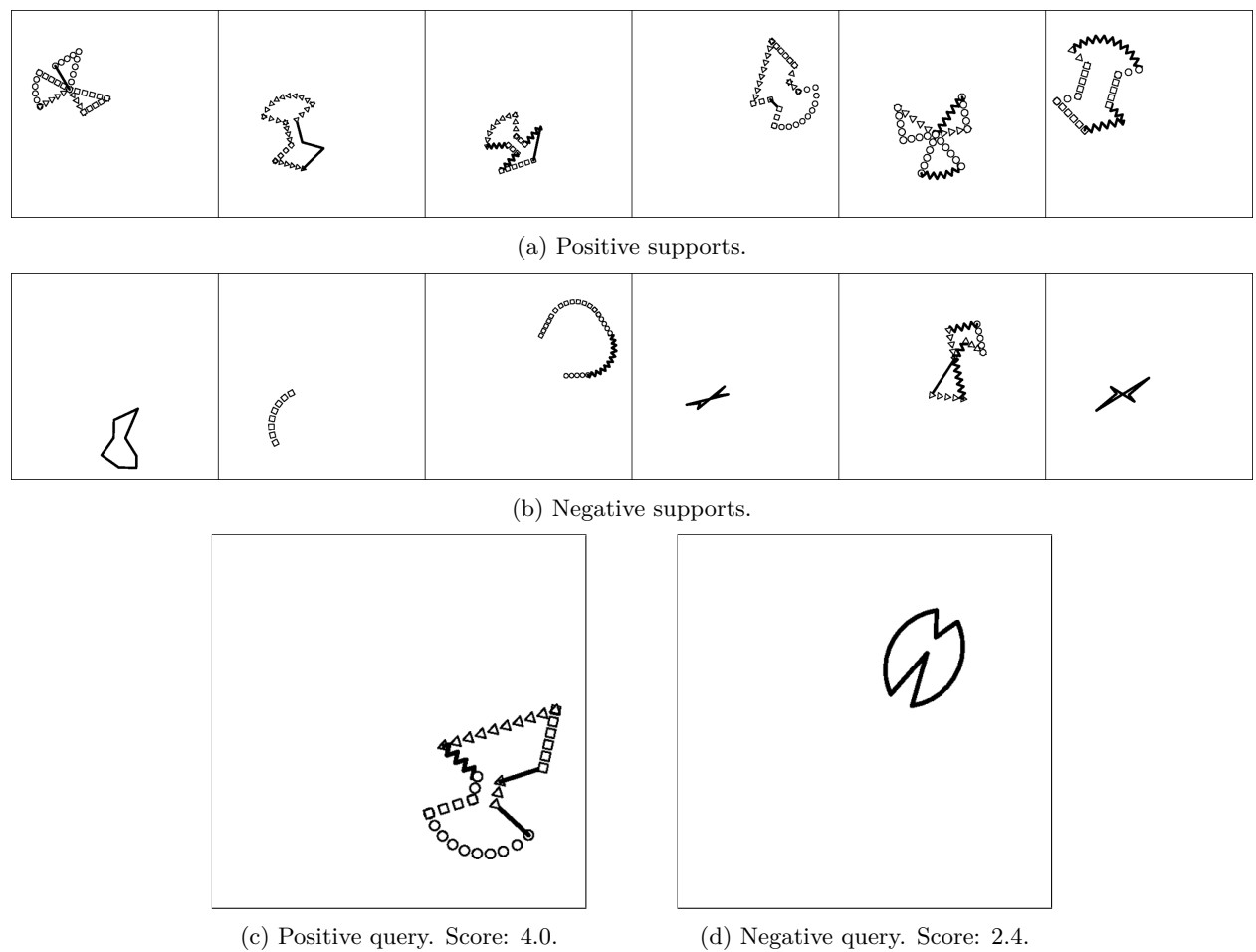

(a) Positive supports.

(b) Negative supports.

(c) Positive query. Score: 4.0.   (d) Negative query. Score: 2.4.

Figure 27: **Bongard-Logo Novel Abstract Shape: Correct for positive, incorrect for negative**.

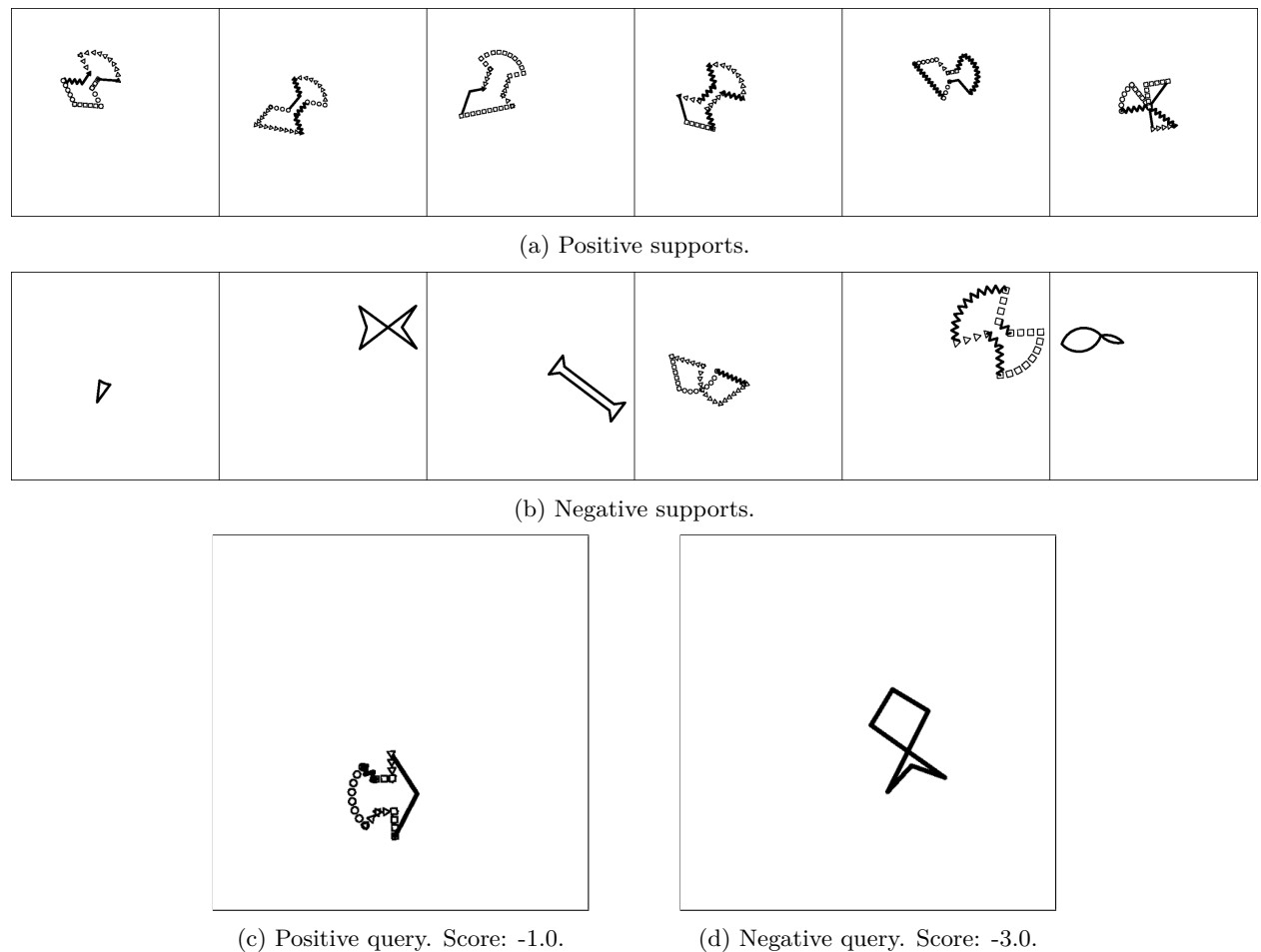

(a) Positive supports.

(b) Negative supports.

(c) Positive query. Score: -1.0.

(d) Negative query. Score: -3.0.

Figure 28: **Bongard-Logo Novel Abstract Shape: Incorrect for positive, correct for negative**.

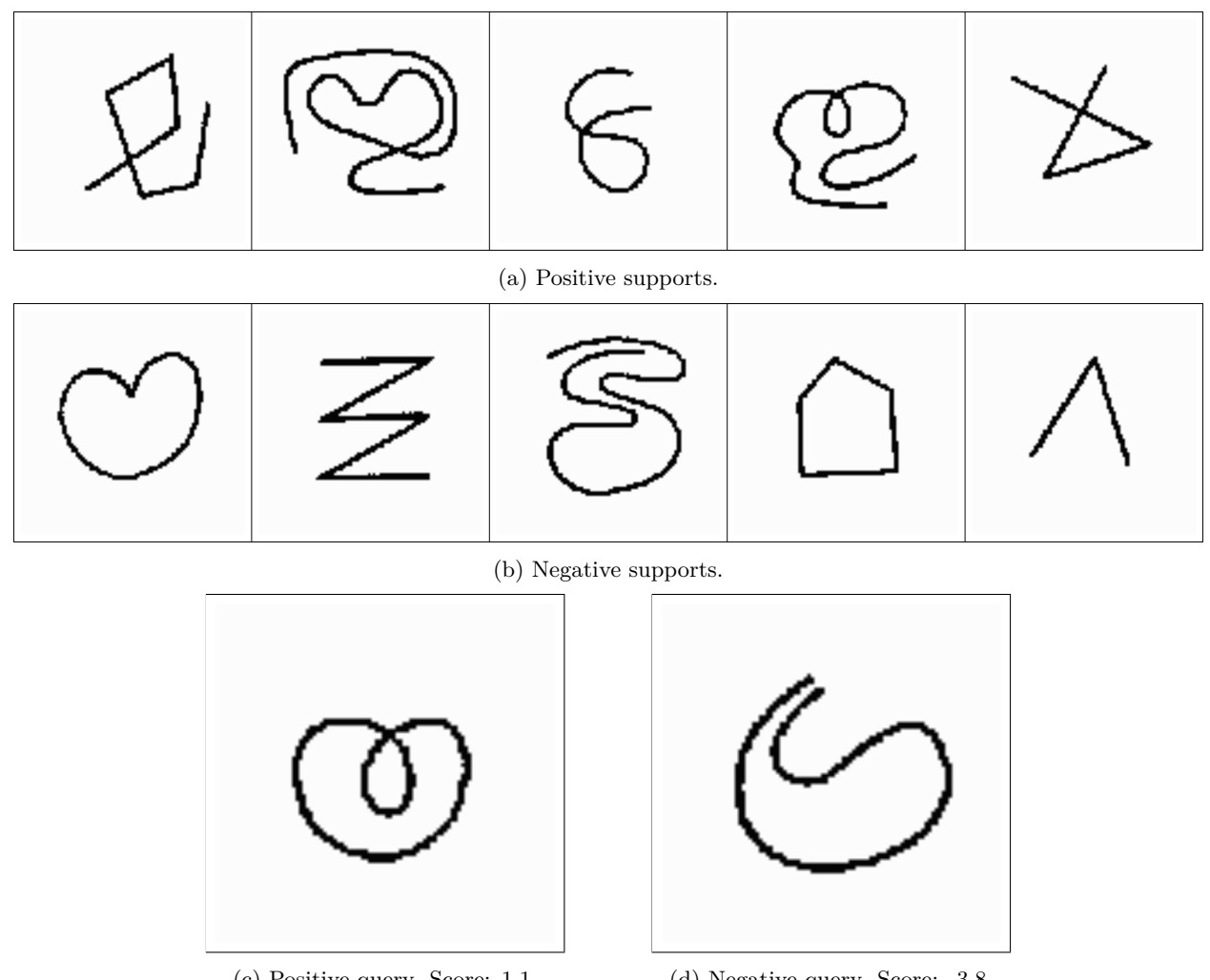

(a) Positive supports.

(b) Negative supports.

(c) Positive query. Score: 1.1.

(d) Negative query. Score: -3.8.

Figure 29: **Bongard-Classic: Correct guess for both queries**.

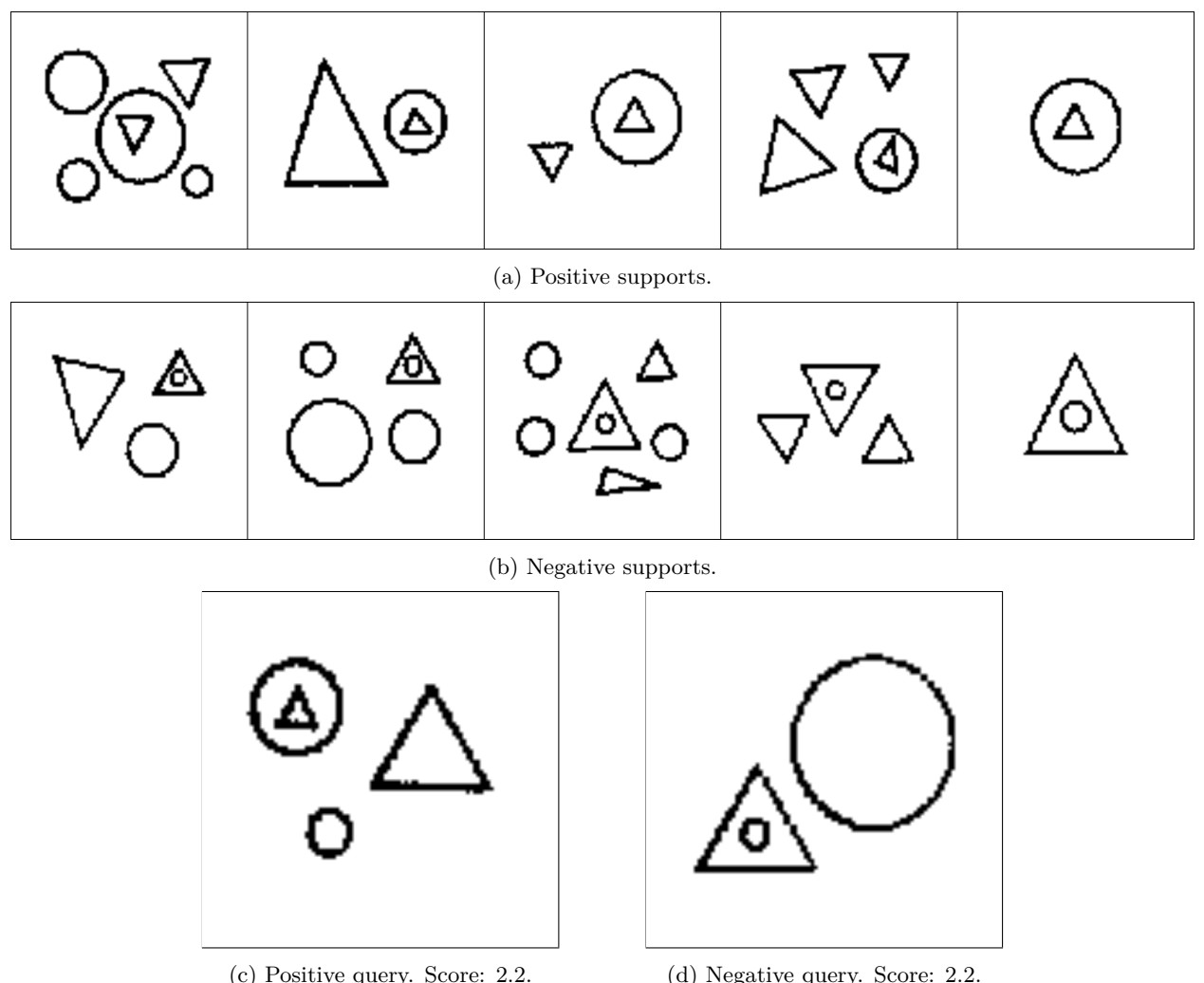

(a) Positive supports.

(b) Negative supports.

(c) Positive query. Score: 2.2.    (d) Negative query. Score: 2.2.

Figure 30: **Bongard-Classic: Correct for positive, incorrect for negative**.

