# OpenReview forum: "Support-Set Context Matters for Bongard Problems"
_TMLR — Accepted by TMLR_

### Review · Reviewer_3X8C · 2024-07-03

**Summary Of Contributions:**

This work proposes shows that simple feature normalization can help improve significantly the performance of different baselines on Bongard Problems. Moreover, a transformer architecture is applied on top of the (normalized) features to predict either some "class prototypes" or a separating hyperplane to perform the binary classification of the Bongard problem. They observe on several experiments that both the feature normalization and the transformer "head" on top of the features can boost performance.

**Audience:**

Yes

**Broader Impact Concerns:**

No concerns

**Claims And Evidence:**

Yes

**Requested Changes:**

**Required changes for acceptance:** Discuss the domain gap between natural images and Bongard-LOGO in section 4.6 when comparing to GPT-4V or evaluate on Bongard-HOI (see the above section).

**Optional changes to improve the paper:**
- Evaluate GPT-4V on Bongard-HOI
- Apply feature normalization to some existing baselines to see if they also benefit from it.
- Re-name the "feature adaptation" to something more descriptive like "transformer-based classifier head" (just a suggestion).

**Update after rebuttal:** After having read other reviews and authors comments, I would advise this paper to be accepted at TMLR.

**Strengths And Weaknesses:**

**Strength**

Paper is well written and does a good job at introducing the reader to the Bongard Problems and existing approaches. Moreover, the results are encouraging and the comparison to prior work on Bongard problems is quite extensive.

**Weakness**

There is however a comparison which did not seem entirely fair, which is that of GPT4-V. Authors state in section 3.2. that due to the particular nature of the Bongard-LOGO images (i.e. simple line drawings with abstract shapes), the features of a CLIP backbone did not yield good results, thus they trained a backbone from scratch exclusively on Bongard-LOGO images. It is reasonable to assume that GPT4-V will have been trained on a dataset similar to CLIP (with mostly real images) and thus the Bongard-LOGO would also present a strong domain shift. I do agree that if the ultimate goal is to assess AGI capabilities of a model, then Bongard-LOGO could be an interesting benchmark, however, I think it is quite reasonable to assume that the performance of GPT4-V (and in general most large pre-trained VLM's) would be much better on Bongard-HOI than Bongard-LOGO.

I do not think authors necessarily need to compare with VLM's for this paper to be interesting but with the decision to only compare on Bongard-LOGO the authors seems to conclude that autoregressive VLM's are still far from current SOTA without discussing the domain shift and I think experiments on Bongard-HOI would have been very interesting.

I can also understand the authors' point that GPT-4V is expensive to evaluate as it is behind a paywall, but there are other open source VLM's such as LLAMA wich could be used.

I also found confusing the naming of the method "Feature adaptation" to refer to the transformer head that predicts the prototypes / hyperplane from the visual feaures. To my understanding the hyperplane or prototypes will be used to classify based on the "original" input features to the transformer. It is therefore not adapting the features themselves. Initially I expected the transformer would output a transformed version of the features but the output is directly a "classifier".

Finally, I think feature standarization could be applied to several (if not all) of the baselines in tables 1 and 2. It would also have been interesting to see how it would affect the performance of other baselines.

---

> ### Author Response · Authors · 2024-08-02
> **Response to Review**
>
> Thank you for the thorough review. We are glad that you found the paper “well-written,” the results “encouraging” and the comparison to prior works “extensive.”
>
> Here are are our responses to your suggestions:
> - “**Discuss the domain gap between natural images and Bongard-LOGO in section 4.6 when comparing to GPT-4V or evaluate on Bongard-HOI.**” This is a great point, and we have added a discussion of the domain gap to the paper. Although we did not evaluate GPT-4V on Bongard-HOI due to the prohibitive cost, we do have a small comparison of our method on Bongard-HOI to MMICL, which is a recent open-source autoregressive VLM built upon InstructBLIP. Although MMICL performs much better on Bongard-HOI than GPT-4V performs on Bongard-LOGO, likely due to the domain shift between Bongard-LOGO and GPT-4V training data, it still performs worse than our best-performing model. It appears that even on Bongard-HOI our simple methods are useful in comparison to recent autoregressive vision-language models.
> - “**I also found confusing the naming of the method "Feature adaptation" to refer to the transformer head that predicts the prototypes / hyperplane from the visual features.**”: We agree that this is confusing. In our updated paper, we call the Transformer approach only “support-set Transformer” and use the term “in-context feature adaptation” to refer to the standardization method only. When referring to both approaches together, we use the term “support-set context.” We have also modified the paper title to reflect this change.
> - “**Finally, I think feature standardization could be applied to several (if not all) of the baselines in tables 1 and 2. It would also have been interesting to see how it would affect the performance of other baselines.**” Thank you for the suggestion. We agree that this is a very interesting set of experiments to try. It is worth noting that many of the published baselines we included (without feature normalization) closely resemble the new baselines we created. For instance, ProtoNet shares similarities with our Prototype baseline, as both techniques involve computing class prototypes and classifying query images based on their distance from each prototype.
>
> Please let us know if we misunderstood any of your original comments, or if you have new comments to add. We will be grateful for any additional feedback, and will work to respond before the discussion period ends.

---

### Review · Reviewer_aq8m · 2024-07-09

**Summary Of Contributions:**

This manuscript addresses the challenge of solving Bongard problems. The authors propose adapting image features based on the entire data set, rather than relying on individual data. To achieve this, they introduce a student-teacher framework, where an SVM model provides guidance to a Transformer encoder. This approach leads to significant improvements in accuracy on the Bongard-LOGO and Bongard-HOI benchmarks, surpassing the best existing methods.

**Audience:**

Yes

**Broader Impact Concerns:**

The authors should provide a Broader Impact Statement.

**Claims And Evidence:**

No

**Requested Changes:**

Please see Weaknesses for details about recommendation for acceptance.

**Strengths And Weaknesses:**

[Strengths]

1.The manuscript introduces an innovative student-teacher framework that leverages the strengths of both SVM and Transformer encoder models.

2.The proposed methods demonstrate substantial improvements in accuracy on benchmark datasets Bongard-LOGO and Bongard-HOI.

[Weaknesses]

1.Section 3.4: The assumption that the mean of the union of two sets represents the shared context of positive and negative sets lacks theoretical basis and is not experimentally validated by the authors. It is unclear whether the standardization operation really removes the shared context. The authors are supposed to conduct experiments to verify this assumption.


2.The symbols in Equations *(2) and (3) are not clearly defined. It is unclear what $\hat{p}$, $\hat{n}$, and $\hat{h}$ represent, whether they are vectors, and how they are obtained. The source of ground-truth $p$, $n$, and $h$ is also not specified. The authors should clarify these definitions and provide details on how these values are derived.

3.The authors propose fitting the Transformer encoder to the SVM classification boundary but do not explain why the SVM classification boundary itself is not used directly. In knowledge distillation, the teacher model is typically more complex, containing richer knowledge, while the student model is simpler. Using a more powerful Transformer encoder to learn SVM knowledge might lead to overfitting. The authors should discuss the rationale behind this approach.

4.The authors use random flipping of positive/negative indicators for some supports to prevent overfitting. They should provide experimental evidence demonstrating the effectiveness of this technique in avoiding overfitting.

5.The claim “We selected our student-teacher approach as it focuses the model on rule-making” lacks theoretical or experimental support. The authors should include relevant theoretical foundations or experimental analysis to substantiate this claim.

---

> ### Author Response · Authors · 2024-08-02
> **Response to Review, Part 1**
>
> Thank you for your thoughtful review. We are pleased that you found the approach interesting and appreciated the improvements in accuracy over prior methods. We are very grateful for your helpful points of feedback and address each of them below:
>
> - “**Section 3.4: The assumption that the mean of the union of two sets represents the shared context of positive and negative sets lacks theoretical basis. It is unclear whether the standardization operation really removes the shared context.**” We agree that the assumption in the original submission lacked evidence. The paper references the Bongard problem in Figure 1 and describes how considering context may allow us to ignore the laptop present in all positive and negative images. A human solving this problem may indeed recognize and choose to ignore any shared semantics. However, our per-feature standardization (mean and variance computed for each feature separately) only removes shared semantics if there are some similar dimensions in the positive and negative feature vectors and if these similar dimensions reflect semantics about the problem. This depends on how the vision backbone has learned to organize the feature space. We have modified Section 3.4 to reflect the fact that this is just a high-level motivation for support-set standardization but is not a claim.
> - “**The symbols in Equations (2) and (3) are not clearly defined.**” Thank you for the suggestion. The $p$, $\hat{p}$, $n$, $\hat{n}$, $h$, and $\hat{h}$ are all vectors. $\hat{p}$, $\hat{n}$, and $\hat{h}$ are the Transformer output tokens that correspond to the learnable “task” input tokens. $p$ and $n$ are prototypes computed by averaging the positive and the negative examples in a Bongard problem, respectively. The hyperplane $h$ is computed by fitting an SVM to a Bongard problem and extracting the corresponding hyperplane decision boundary. We have added this explanation to the updated paper.
> - “**The authors propose fitting the Transformer encoder to the SVM classification boundary but do not explain why the SVM classification boundary itself is not used directly.**” The paper contains baselines where the SVM classification boundary is used directly without incorporating any Transformer or additional learning. These are labeled as "SVM" and "SVM + standardize" in the tables (see, e.g., Tables 1 and 2). These SVM baselines tend to perform worse than SVM-Mimic, the Transformer version. The motivation behind using a Transformer is that the Transformer (1) is another way to incorporate support-set-level knowledge and (2) can learn priors that generalize across Bongard problems, whereas the SVM baseline is fit for each Bongard problem independently and therefore learns no such priors. Please let us know if there is another experiment that you have in mind for strengthening the paper.
> - “**In knowledge distillation, the teacher is typically more complex than the student. Using a more powerful Transformer encoder to learn SVM knowledge might lead to overfitting. The authors should discuss the rationale behind this approach.**” We agree that the use of a “simple” teacher is unconventional, but we believe that is part of what makes the method interesting. Similar to traditional student-teacher setups, the “teacher” in our method—the SVM—solves a much easier task than the student. Namely, the student SVM is fit using all available examples in a Bongard problem, while the teacher Transformer sees a subset of the examples. Additionally, in the case of Bongard-HOI, we incorporate label noising. Through this unequal setup, the Transformer learns how to cope with incomplete or incorrect information. We do not find that the Transformer suffers from overfitting. If the Transformer were to overfit, SVM-Mimic (Transformer method) would achieve worse performance than the SVM baseline, but instead we see that it achieves better performance (see, e.g., Tables 1 and 2). This implies that the Transformer can learn valuable information beyond what is explicitly provided by the SVM. We have added this more detailed discussion to the paper in section 3.5.
> - “**[The authors] should provide experimental evidence demonstrating the effectiveness of [random flipping] in avoiding overfitting.**” Thank you for the suggestion. We have experimental evidence that shows that train-time random flipping (called label noise in the paper) leads to higher accuracy, which suggests that it prevents overfitting. Trained without random flipping, SVM-Mimic attains $72.08\pm0.10$ accuracy on Bongard-HOI. Trained with random flipping, it attains $72.45\pm0.16$ accuracy, which is a small but statistically significant improvement. Each of these values is averaged over three trained models.

---

> > ### Author Response · Authors · 2024-08-02
> > **Response to Review, Part 2**
> >
> > - “**The claim ‘We selected our student-teacher approach as it focuses the model on rule-making’ lacks theoretical or experimental support.**” We agree that this sentence is unclear and have removed it. What we intended to explain was the following: given that the task of a Bongard problem is to produce some rule that divides the positive and negative examples, and given that a “rule” is equivalent to a hyperplane in the embedding space, we supervise our Transformers to output hyperplane decision boundaries, either directly, or in the form of class prototypes (which admit a decision boundary). This is as opposed to supervising a cross-entropy loss on the classification task. In other words, rather than saying “our method focuses the model on rule-making”, we should have said “our Transformers output hyperplane decision boundaries.”
> > - “**The authors should provide a Broader Impact Statement.**” Thank you for the suggestion. We have included a Broader Impact statement in the updated paper.
> >
> > Please let us know if we misunderstood any of your original comments, or if you have new comments to add. We will be grateful for any additional feedback, and will work to respond before the discussion period ends.

---

### Review · Reviewer_LhM4 · 2024-07-23

**Summary Of Contributions:**

The paper addresses the challenges in solving Bongard problems, which involve deriving an abstract concept from a set of positive and negative support images to classify new query images. The authors propose that the low accuracy of current methods is due to the lack of adaptation of image features based on the support set as a whole. They introduce two main contributions: a simple parameter-free technique called support-set standardization, and a Transformer-based approach for learning set-level knowledge. These methods lead to new state-of-the-art accuracies on Bongard-LOGO and Bongard-HOI datasets, significantly improving performance over existing approaches with equivalent vision backbone architectures.

**Audience:**

Yes

**Claims And Evidence:**

Yes

**Requested Changes:**

Please see above

**Strengths And Weaknesses:**

Strengths

- The proposed methods are simple and easy to follow.
- The paper seems to provide a comprehensive comparison with existing methods.

Weaknesses

I must admit that I am not an expert in this specific domain, so it's hard for me to fully assess the novelty and potential impact of this paper's contributions. The background involving IQ tests is interesting, I wonder whether this is a niche topic and how it differs from other few-shot learning problems and its uniqueness and potential implications for broader fields. The paper notes that irregular sample distances greatly affect accuracy, with standardization providing substantial benefits, which reminds me of metric learning efforts that aim to control variance among representations. However, the paper doesn't compare its methods with existing work in metric learning. A detailed discussion and comparison with these techniques would be helpful. Additionally, some methodological choices are not well-explained. For instance, the authors mention, "we find that CLIP performs poorly, and we therefore train a ResNet," but don't provide a detailed rationale. The authors should elaborate on these choices to improve the clarity and robustness of their methodology.

---

> ### Author Response · Authors · 2024-08-02
> **Response to Review**
>
> Thank you for the thoughtful review. We are glad to hear that you found the methods “simple and easy to follow” and the paper “comprehensive.” Below, we have responded to your points.
>
> - “**I wonder whether this is a niche topic and how it differs from other few-shot learning problems and its uniqueness and potential implications for broader fields.**” This is indeed a niche topic, but in our opinion it is important and interesting due to its relevance to improving human-like reasoning in models. Our methods are likely applicable to many other few-shot learning problems. In particular, support-set standardization is likely widely applicable due to its simplicity.
> - “**A detailed discussion and comparison with [metric learning] techniques would be helpful.**” While we did not compare extensively against metric learning baselines, we did experiment with contrastive learning, a form of metric learning, both in our approach and in the baselines. We have numerous experiments that use CLIP, which was trained with a contrastive objective. We also trained an encoder from scratch for Bongard-LOGO with a contrastive loss. This contrastive loss encourages positive images in a Bongard-LOGO problem to be close to each other in the embedding space and far from all negative images in the same problem. Additionally, note that some published baselines have connections to metric learning—for example, ProtoNet enforces that each feature vector is close to its correct class prototype and far from the incorrect class prototype(s). We have added a note about metric learning methods in section 3.2.
> - “**Additionally, some methodological choices are not well-explained. For instance, the authors mention, ‘we find that CLIP performs poorly, and we therefore train a ResNet,’ but don't provide a detailed rationale.**” Thank you for pointing this out. The rationale is that CLIP is trained on Internet images—primarily natural images—whereas all Bongard-LOGO images are synthetic images (line drawings). Note that we experimented with the CLIP backbone on Bongard-LOGO (this is the TPT baseline in Table 1) and saw that it performed much worse than all baselines that use our custom encoder. We have now clarified this detail in the paper.
>
> Please let us know if we misunderstood any of your original comments, or if you have new comments to add. We will be grateful for any additional feedback, and will work to respond before the discussion period ends.

---

### Decision · Action_Editor_PeDB · 2024-09-12

**Recommendation:** Accept as is

**Comment:**

Reviewers are generally recommending acceptance and the AE agrees. The feature adaptation method presented is interesting and novel possibly even beyond the context of Bongard problems. Paper is well written and related work extensive.

**Audience:**

Although the target audience for this paper is narrow, the problem is interesting, the method novel and could possibly be useful beyond Bongard problems.

**Claims And Evidence:**

Claims are properly supported by convincing evidence.